# ZERO-ORDER SHARPNESS-AWARE MINIMIZATION

## ABSTRACT

Prompt learning has become a key method for adapting large language models to specific tasks with limited data. However, traditional gradient-based optimization methods for tuning prompts are computationally intensive, posing challenges for efficiency. We introduce ZOSA (Zero-Order Sharpness-Aware Minimization), a novel optimization framework that integrates zero-order optimization with sharpness-aware minimization to enhance prompt tuning. ZOSA employs Rademacher perturbation vectors to estimate gradients without requiring backpropagation. By incorporating sharpness-aware principles, it targets flat minima in the loss landscape, improving generalization. An adaptive learning rate, guided by loss variability, further ensures stable convergence. Experiments on few-shot learning tasks, such as text classification and natural language inference, show that ZOSA significantly outperforms existing methods. With its theoretical foundation and computational efficiency, ZOSA offers a practical solution for prompt-based learning in resource-limited settings.

## 1 INTRODUCTION

Zeroth-order (ZO) optimization has become indispensable in machine learning scenarios where gradient information is inaccessible or computationally prohibitive, such as black-box adversarial attacks (Ru et al., 2020; Hiranandani et al., 2021) and memory-constrained fine-tuning of large language models (LLMs) (Malladi et al., 2023b; Zhang et al., 2024b). Unlike first-order methods that rely on backpropagation, ZO algorithms estimate gradients solely through function evaluations, enabling applications in resource-limited environments (Liu et al., 2018b; Chen et al., 2019; Shu et al., 2024).

However, traditional ZO methods suffer from high variance in gradient estimates, leading to slow convergence and suboptimal generalization, particularly in high-dimensional non-convex landscapes (Chen et al., 2019; Nazari et al., 2020). Adaptive ZO optimizers, such as ZO-AdaMM (Chen et al., 2019; Nazari et al., 2020), attempt to mitigate these issues by incorporating moment estimates for better scaling of updates. Yet, they underutilize historical information, resulting in noisy estimates and limited performance gains (Shu et al., 2025). Recent advances like R-AdaZO (Shu et al., 2025) refine moment utilization through variance reduction on first-moment estimates and improved second-moment approximations, achieving faster convergence. Meanwhile, sharpness-aware minimization (SAM) (Foret et al., 2021) has emerged as a powerful technique in first-order settings to enhance generalization by seeking flat minima, but its direct application to ZO is challenging due to the absence of gradients. SABO (Ye et al., 2024) extends SAM to zero-order optimization by reparameterizing objectives over Gaussian distributions and approximating sharpness-aware updates via stochastic gradients. Similarly, FZOO (Dang et al., 2025) accelerates ZO fine-tuning of LLMs using batched one-sided estimates and adaptive step-sizes based on loss standard deviations, approaching Adam-like speeds with inference-level memory.

Despite these progresses, existing ZO methods often trade off between memory efficiency, convergence speed, and generalization. For instance, MeZO (Malladi et al., 2023a) reduces memory to inference levels but requires significantly more forward passes than Adam. To bridge this gap, we introduce Zeroth-Order Sharpness-Aware (ZOSA) optimization, a novel adaptive ZO optimizer that integrates sharpness-aware mechanisms with refined variance reduction and adaptive scaling. ZOSA employs batched Rademacher perturbations for efficient one-sided gradient estimates, computes adaptive step-sizes using the standard deviation of batch losses, and incorporates a SAM-like perturbation at a scaled point (with effective radius $\rho/\epsilon$) to promote flat minima and better general-

ization. This design reduces variance in estimates (building on R-AdaZO (Shu et al., 2025)) while minimizing forward passes. Our contributions are summarized as follows:

- We propose ZOSA, an efficient ZO optimizer that seamlessly combines sharpness-aware updates with adaptive loss-variance-based scaling, with minimal inference-time memory overhead. The algorithm features a simple design, requiring no additional updates to a $\Sigma$ matrix (thus avoiding the extra storage burden and computational complexity associated with $\Sigma$ matrix updates), it offers fast computation, significantly fewer function queries, and accuracy superior to or on par with SABO, which is the state-of-the-art sharpness-aware ZO algorithm, providing a practical and easy-to-implement solution for ZO optimization.
- We provide comprehensive theoretical analysis, including proofs of approximate equivalence to a normalized SAM rule with effective sharpness radius $\rho/\epsilon$, rigorous variance reduction bounds, and convergence guarantees under standard assumptions, further solidifying its theoretical foundations.
- Extensive experiments on synthetic problems and LLM fine-tuning tasks demonstrate ZOSA's superiority in convergence speed, resource efficiency, and performance stability, highlighting its potential in real-world applications.

## 2 RELATED WORKS

ZO optimization research primarily advances in gradient estimation and update rules, with growing emphasis on adaptivity, generalization, and applications to large-scale models like LLMs. Early ZO methods rely on finite-difference approximations for gradients, such as two-point estimates (Liu et al., 2018b;a). To address high-dimensional challenges, random direction sampling (e.g., Gaussian, Rademacher, or coordinate-wise) reduces query complexity while maintaining unbiased estimates (Chen et al., 2019; Shu et al., 2024; Nesterov & Spokoiny, 2017). Recent works like MeZO (Malladi et al., 2023a) apply these techniques to LLM fine-tuning, replacing backpropagation with forward passes to achieve inference-level memory usage. However, MeZO's fixed step-sizes lead to slow convergence, often requiring 10-20× more iterations than first-order methods, as practitioners highlight its dependence on local Hessian rank rather than parameter count.

**ZO for LLM Fine-Tuning**. With the rise of LLMs, ZO has been tailored for memory-efficient fine-tuning in resource-constrained settings. For instance, DP-ZO (Lin et al., 2024) introduces differential privacy into ZO for private LLM adaptation using forward-only perturbations. Quantized variants like QuZO (Qu et al., 2025) enable low-bit ZO fine-tuning, achieving performance comparable to MeZO on tasks like GLUE while reducing computational overhead. HiZOO (Cai et al., 2024) leverages diagonal Hessian approximations to enhance ZO updates, and FedMeZO (Zhang et al., 2024a) extends this to federated learning, proving convergence in distributed settings. SubZero (Chen et al., 2024a) tackles sparsity by optimizing in random subspaces, mitigating dimensionality curses in billion-parameter models. These methods collectively demonstrate ZO's potential for LLM adaptation but often overlook generalization in non-convex landscapes (Zhang et al., 2024b; Malladi et al., 2023b).

**Adaptive ZO Optimizers and Variance Reduction**. Adaptive methods improve upon basic SGD-like ZO by incorporating momentum and scaling. ZO-AdaMM (Chen et al., 2019; Nazari et al., 2020) adapts Adam's moments to ZO but suffers from high-variance estimates in noisy environments. R-AdaZO (Shu et al., 2025) overcomes this by providing the first analysis of variance reduction via first-moment estimates and refining second moments for better geometry capture, yielding faster convergence than ZO-AdaMM. FZOO (Dang et al., 2025) accelerates ZO through batched Rademacher perturbations and adaptive step-sizes based on batch loss standard deviations, emulating normalized-SGD (You et al., 2019) without momentum costs, and seamlessly integrates with PEFT techniques like LoRA (Hu et al., 2021) for further memory savings. Additional variance reduction approaches, such as LOZO (Chen et al., 2024b), use low-rank gradient estimators to capture low-dimensional structures in LLM loss landscapes (Zhou et al., 2025b).

**Sharpness-Aware and Zero-Order Optimization**. SAM (Foret et al., 2021) promotes generalization in first-order optimization by seeking flat minima through neighborhood maximization, but it requires gradients. Extensions like VS-SAM (Liu et al., 2023) suppress variance in perturbations for stable training. In zero-order settings, SABO (Ye et al., 2024) reparameterizes objectives over Gaussian distributions to approximate sharpness-aware stochastic gradients, with proven convergence and

generalization bounds. SharpZO (Yang et al., 2025) hybridizes this for vision-language models, enhancing ZO prompt optimization. These align with empirical findings that flat minima correlate with better performance (Dziugaite & Roy, 2017; Petzka & Sminchisescu, 2021; Andriushchenko & Flammarion, 2022). In LLM contexts, sharpness-aware ZO could alleviate overfitting in zero-order tuning (Sun et al., 2022b; 2023), though existing methods like SAM variants (Huang et al., 2024) are limited to gradient-based scenarios.

ZOSA builds on these foundations by fusing variance-reduced adaptive scaling (Shu et al., 2025), efficient batched estimation, and sharpness-aware updates (Ye et al., 2024), while incorporating elements from recent LLM-specific ZO works (Lin et al., 2024; Cai et al., 2024), to provide a unified, memory-efficient framework for generalizable optimization.

## 3 METHODOLOGY

In the following, we first briefly review the preliminaries of classical zeroth-order gradient estimation (Section 3.1). We then present the motivation and complete workflow of our ZOSA optimizer (Section 3.2), and finally offer a theoretical analysis of the gradient estimation properties (Section 3.3).

### 3.1 PRELIMINARIES

We consider the standard supervised fine-tuning (or prompt tuning) objective on a labeled dataset $\mathcal{D} = \{(x_i, y_i)\}_{i=1}^{|\mathcal{D}|}$: $L(\theta; \mathcal{B}) = \frac{1}{|\mathcal{B}|} \sum_{(x,y) \in \mathcal{B}} \ell(h(\theta; x), y)$, where $\theta \in \mathbb{R}^d$ represents the trainable parameters, and $h(\cdot; x)$ denotes the frozen pretrained LLM with appended prompts.

**Classical ZO Gradient Estimation** Given a perturbation radius $\epsilon > 0$ and $z \in \mathbb{R}^d$ sampled as $z \sim \mathcal{N}(0, I_d)$, where $I_d \in \mathbb{R}^{d \times}$ is the identity matrix of dimension $d$, the Classical ZO estimates the gradient on $\mathcal{B}$ via:

$$\hat{\nabla} L(\theta; \mathcal{B}) = \frac{L(\theta + \epsilon z; \mathcal{B}) - L(\theta - \epsilon z; \mathcal{B})}{2\epsilon} z \approx z z^\top \nabla L(\theta; \mathcal{B}). \tag{1}$$

Averaging over $N$ i.i.d. draws $\{z_i\}_{i=1}^N$ yields the $N$-ZO estimator $\hat{\nabla}_N L = \frac{1}{N} \sum_{i=1}^N \hat{\nabla}_i L$.

**From Classical ZO to ZO-SGD.** Let $\theta_t \in \mathbb{R}^d$ denote the trainable parameters at iteration $t$, $\mathcal{B}_t$ be the mini-batch sampled at iteration $t$, and $\hat{g}_t := \hat{\nabla} L(\theta_t; \mathcal{B}_t)$ be the zeroth-order gradient estimator computed on $\mathcal{B}_t$. Replacing the back-propagation gradient in SGD with this zeroth-order estimate directly yields the zeroth-order stochastic update

$$\theta_{t+1} = \theta_t - \eta_t \hat{g}_t,$$

where $\eta_t > 0$ is the learning rate. MeZO realizes this update in-place with the memory tricks above and serves as a baseline for improvements. The introduction of Fast Zeroth-Order Optimizer (FZOO) is in Appendix A.1.

### 3.2 MOTIVATION OF ZOSA

To enhance generalization while maintaining efficiency, ZOSA builds upon FZOO by incorporating SAM principles. This integration targets flatter minima in the loss landscape. Specifically, ZOSA first estimates the gradient at the current parameters using batched Rademacher perturbations, computes an adaptive perturbation based on the estimated gradient normalized by loss variance, and then performs a second gradient estimation at the perturbed point for the final update. This dual-estimation approach combines FZOO's memory-efficient zeroth-order strategy with SAM-like sharpness awareness, normalized via variance for stability in high-dimensional spaces.

To obtain a low-variance zeroth-order gradient estimate, we draw $m$ independent Rademacher random vectors $u_1, \ldots, u_m \overset{\text{i.i.d.}}{\sim} \mathbb{R}^d$ (each component of $u_i \in \{-1, +1\}^d$ with equal probability $1/2$, independent across coordinates and samples). For a fixed perturbation radius $\epsilon > 0$, we perform $m + 1$ forward passes to query the following scalar function values:

$$l_0 = L(\theta_t; \mathcal{B}_t), \tag{2}$$

$$l_i = L(\theta_t + \epsilon u_i; \mathcal{B}_t), \quad i = 1, \ldots, m. \tag{3}$$

---

**Algorithm 1** ZOSA (Zero-Order Sharpness-Aware) Optimizer

---

**Require:** Model parameters $\theta$, loss function $L(\theta)$, downstream labeled dataset $\mathcal{D}$, batch size $|\mathcal{B}|$, sharpness-aware radius $\rho$, perturbation scale $\epsilon$, number of sampled directions $m$, learning rate $\eta$
**Ensure:** Optimized model parameters $\theta$
1: Initialize $\theta$
2: **while** not converged **do**
3:      Sample mini-batch $\mathcal{B} \subset \mathcal{D}$ uniformly at random
4:      **Estimate gradient at the original point:**
5:      Generate $m$ Rademacher perturbation vectors $u_i$ for $i = 1$ to $m$
6:      Compute original loss $l_0 = L(\theta; \mathcal{B})$
7:      **for** each $u_i$ **do**
8:          Perturb parameters: $\theta' = \theta + \epsilon \cdot u_i$
9:          Compute perturbed loss $l_i = L(\theta')$
10:         Restore parameters: $\theta' = \theta$
11:     **end for**
12:     Estimate gradient $g_t = \frac{1}{m} \sum_{i=1}^{m} \frac{l_i - l_0}{\epsilon} \cdot u_i$
13:     **Sharpness-aware perturbation:**
14:     $\sigma_t = \text{std}([l_0, l_1, \ldots, l_m])$
15:     **if** $\sigma_t > 0$ **then**
16:         Compute perturbation $\epsilon_{\text{sam}} = \rho \cdot \frac{g_t}{\sigma_t + 10^{-8}}$
17:     **else**
18:         $\epsilon_{\text{sam}} = 0$
19:     **end if**
20:     **Estimate gradient at the perturbed point:**
21:     Perturb parameters: $\theta_{\text{pert}} = \theta + \epsilon_{\text{sam}}$
22:     Generate $m$ new Rademacher perturbation vectors $u_{\text{pert},i}$ for $i = 1$ to $m$
23:     Compute perturbed point loss $l_{\text{pert}} = L(\theta_{\text{pert}}; \mathcal{B})$
24:     **for** each $u_{\text{pert},i}$ **do**
25:         Perturb parameters: $\theta'_{\text{pert}} = \theta_{\text{pert}} + \epsilon \cdot u_{\text{pert},i}$
26:         Compute perturbed loss $l_{i,\text{pert}} = L(\theta'_{\text{pert}})$
27:         Restore parameters: $\theta'_{\text{pert}} = \theta_{\text{pert}}$
28:     **end for**
29:     Estimate gradient $g_{\text{pert}} = \frac{1}{m} \sum_{i=1}^{m} \frac{l_{i,\text{pert}} - l_{\text{pert}}}{\epsilon} \cdot u_{\text{pert},i}$
30:     **Compute adaptive learning rate at perturbed point:**
31:     $\sigma_{t,\text{pert}} = \text{std}([l_{\text{pert}}, l_{1,\text{pert}}, \ldots, l_{m,\text{pert}}])$
32:     Set adaptive learning rate $\eta_{\text{adaptive}} = \begin{cases} \frac{\eta}{\sigma_{t,\text{pert}} + 10^{-8}} & \text{if } \sigma_{t,\text{pert}} > 0 \\ \eta & \text{otherwise} \end{cases}$
33:     **Update parameters:**
34:     $\theta = \theta - \eta_{\text{adaptive}} \cdot g_{\text{pert}}$
35:     Restore parameters to original point: $\theta = \theta - \epsilon_{\text{sam}}$
36: **end while**
37: **return** $\theta$

---

The batched one-sided Rademacher gradient estimator at the original point $\theta_t$ is then constructed as

$$\hat{g}_t = \frac{1}{m} \sum_{i=1}^{m} \frac{l_i - l_0}{\epsilon} u_i = \frac{1}{m\epsilon} \sum_{i=1}^{m} (l_i - l_0) u_i. \tag{4}$$

This estimator is unbiased for the directional derivative along each $u_i$ and exhibits significantly lower variance than single-sample ($m = 1$) estimates.

The estimated variance $\sigma_t^2$ at the original point is computed as:

$$\sigma_t^2 = \frac{1}{m-1} \sum_{i=1}^{m} \left( l_i - \frac{1}{m} \sum_{j=1}^{m} l_j \right)^2. \tag{5}$$

Next, the sharpness-aware perturbation is calculated as $\epsilon_{\text{sam}} = \rho \frac{g_t}{\sigma_t}$ (if $\sigma_t > 0$, else zero), where $\rho$ is the SAM perturbation radius. We then move to the perturbed point $\theta_t + \epsilon_{\text{sam}}$ and generate a new set

of $m$ i.i.d. Rademacher vectors $u_{\text{pert},1}, \ldots, u_{\text{pert},m}$. Compute $l_{\text{pert},i} = L(\theta_t + \epsilon_{\text{sam}} + \epsilon u_{\text{pert},i}; \mathcal{B}_t)$ and $l_{\text{pert}} = L(\theta_t + \epsilon_{\text{sam}}; \mathcal{B}_t)$. The gradient estimate at the perturbed point $g_{\text{pert}}$ is:

$$g_{\text{pert}} = \frac{1}{\epsilon m} \sum_{i=1}^{m} (l_{\text{pert},i} - l_{\text{pert}}) u_{\text{pert},i}. \tag{6}$$

The estimated variance at the perturbed point $\sigma_{t,\text{pert}}^2$ is:

$$\sigma_{t,\text{pert}}^2 = \frac{1}{m-1} \sum_{i=1}^{m} \left( l_{\text{pert},i} - \frac{1}{m} \sum_{j=1}^{m} l_{\text{pert},j} \right)^2. \tag{7}$$

Our ZOSA updates the parameters (after restoring to the original $\theta_t$) according to:

$$\theta_{t+1} = \theta_t - \eta \frac{g_{\text{pert}}}{\sigma_{t,\text{pert}}}, \tag{8}$$

where $\eta$ is the base learning rate. The detailed implementation of ZOSA is outlined in Alg. 1.

### 3.3 Understanding ZOSA's Gradient Estimation

**Property 3.1** (Batched One-Sided Rademacher Estimator)

For the zeroth-order gradient estimator used in ZOSA,

$$\hat{g}_t = \frac{1}{m} \sum_{i=1}^{m} \hat{g}_{t,i}, \qquad \hat{g}_{t,i} = \frac{L(\theta_t + \epsilon u_i; \mathcal{B}_t) - L(\theta_t; \mathcal{B}_t)}{\epsilon} u_i,$$

where $u_1, \ldots, u_m \overset{\text{i.i.d.}}{\sim} \text{Rademacher}^d$ are independent Rademacher vectors in $\mathbb{R}^d$, and the empirical loss $L(\cdot; \mathcal{B}_t)$ is twice continuously differentiable with $L$-Lipschitz gradients. The second-order Hessian term vanishes exactly due to the odd symmetry of Rademacher vectors, yielding a bias of $O(\epsilon^2)$. The variance calculation gives

$$\mathbb{E}\left[ \|\hat{g}_t - \nabla L(\theta_t; \mathcal{B}_t)\|^2 \right] \leq \frac{(d-1)\|\nabla L(\theta_t; \mathcal{B}_t)\|^2 + O(\epsilon d^2)}{m} + O(\epsilon^4) = O\left( \frac{d}{m} + \epsilon^2 \right).$$

The proof is provided in Appendix A.2.

Using Rademacher vectors instead of Gaussian perturbations offers computational advantages, as perturbations involve only sign flips, enabling efficient batched forward passes via CUDA parallelism. This fuses multiple matrix multiplications into a single kernel, reducing wall-time by a factor proportional to $m$, making ZOSA suitable for high-dimensional LLM fine-tuning.

**Property 3.2** (Adaptive Scaling via Loss Variance) ZOSA employs an adaptive learning rate $\eta/\sigma_{t,\text{pert}}$, where $\sigma_{t,\text{pert}}$ is the standard deviation of the perturbed losses at the perturbed point. This design draws inspiration from normalized-SGD principles, adapting step sizes based on local curvature without the overhead of momentum. The variance of the loss perturbations satisfies:

$$\mathbb{E}[\sigma_t^2] = \epsilon^2 \|\nabla L(\theta_t; \mathcal{B}_t)\|^2 + O(\epsilon^3 d), \tag{9}$$

where $\sigma_t \approx \epsilon \|\nabla L(\theta_t; \mathcal{B}_t)\|$. Thus, dividing by $\sigma_t$ normalizes the gradient estimate, yielding updates akin to:

$$\theta_{t+1} = \theta_t - \frac{\eta}{\sigma_{t,\text{pert}}} \hat{g}_{\text{pert}} \approx \theta_t - \eta \frac{\nabla L(\theta_t + \epsilon_{\text{sam}})}{\|\nabla L(\theta_t + \epsilon_{\text{sam}})\|}. \tag{10}$$

This receives larger steps in flat regions (small $\sigma_t$) and smaller steps in steep regions, mirroring Adam-style adaptivity at inference-level memory cost. It establishes ZOSA's approximate equivalence to normalized-SAM in the zeroth-order domain with effective sharpness radius $\rho/\epsilon$, enhancing convergence speed and stability. Detailed proofs and the approximate equivalence of SAM are provided in Appendix A.3 and Appendix A.5.

**Property 3.3** (Concentration of $\sigma_t$) Assume the loss $L$ is $L$-smooth and $G$-Lipschitz continuous (i.e., $\|\nabla L(\theta)\| \leq G$ for all $\theta$). The perturbations $u_i$ are i.i.d. Rademacher vectors satisfying $\|u_i\|^2 = d$,

and the loss differences $\Delta l_i := l_i - l_0$ are sub-Gaussian with variance proxy $V^2$. For batch size $m \geq \mathcal{O}(\log(1/\delta))$ (with tighter concentration in high $d$ due to CLT), with probability at least $1 - \delta$,

$$|\sigma_t - \epsilon\|\nabla L(\theta_t)\|| \leq \mathcal{O}\left(\epsilon\sqrt{d}/\sqrt{m} + \epsilon^2\sqrt{d} + \sqrt{\frac{V^2\log(1/\delta)}{m}}\right). \tag{11}$$

This implies

$$\mathrm{Var}[\sigma_t] \leq \mathcal{O}\left(\frac{\epsilon^2 G^2 + V^2}{m} + \frac{\epsilon^2\log m}{m}\right), \tag{12}$$

ensuring that the relative variance $\mathrm{Var}[\sigma_t]/\mathbb{E}[\sigma_t]^2$ is small. The detailed proof is provided in Appendix A.4.

## 4 ANALYSIS

### 4.1 CONVERGENCE ANALYSIS OF ZOSA

This section derives the convergence properties of ZOSA, showing that the average squared gradient norm decreases over iterations under certain assumptions.

**Assumption 4.1 (Smoothness).** Suppose that the loss function $L(\theta)$ is $\mathcal{L}$-smooth, i.e., for all $\theta_1, \theta_2 \in \mathbb{R}^d$, it holds that:

$$L(\theta_2) \leq L(\theta_1) + \langle\nabla L(\theta_1), \theta_2 - \theta_1\rangle + \frac{\mathcal{L}}{2}\|\theta_2 - \theta_1\|^2. \tag{13}$$

**Assumption 4.2 (Bounded Variance).**(Dang et al., 2025) The stochastic gradient $\nabla L(\theta; \mathcal{B})$ has bounded variance:

$$\mathbb{E}\|\nabla L(\theta; \mathcal{B})\|^2 \leq \|\nabla L(\theta)\|^2 + \mathcal{V}^2, \tag{14}$$

where $\mathcal{V}^2$ is a constant. The above is a standard assumption for stochastic gradient descent.

$$\mathbb{E}\|\hat{g}_t\|^2 = \frac{N + d - 1}{N}\|\nabla L(\theta_t)\|^2 + \gamma_t, \quad \gamma_t = O(\epsilon), \tag{15}$$

$$\mathbb{E}\sigma_t^2 = \epsilon^2\|\nabla L(\theta_t)\|^2 + \zeta_t, \quad \zeta_t = O(\epsilon^3). \tag{16}$$

This implies that $\sigma_t \approx \epsilon\|\nabla L(\theta_t)\|$, and the effective perturbation magnitude in ZOSA is approximately $\rho$ after normalization (i.e., the division by $\sigma_t$ effectively scales to match the unit gradient direction with radius $\rho$).

**Theorem 4.3 (Convergence of ZOSA).** Assume the objective function $L(\theta)$ is $\mathcal{L}$-smooth and non-convex with a lower bound $L^*$, the loss perturbations satisfy bounded variance ($\mathbb{E}[(l_i - L(\theta))^2] \leq \sigma^2$), and the gradients are bounded by $G$. The update rule is $\theta_{t+1} = \theta_t - \frac{\eta}{\sigma_{t,\text{pert}}}\hat{g}_{\text{pert}}$, where $\hat{g}_{\text{pert}}$ is the zeroth-order gradient estimate at the perturbed point $\theta_t + \epsilon_{\text{sam}}$, with $\epsilon_{\text{sam}} = \rho\frac{\hat{g}_t}{\sigma_t}$ ($\rho > 0$ is the sharpness radius), $m$ is the number of queries per estimate (Rademacher perturbations), and $\epsilon$ is the perturbation scale. Choose $\eta_t \leq \frac{m}{16d\mathcal{L}}, \epsilon \leq \frac{1}{\sqrt{d\mathcal{L}}}, and \rho \leq \frac{1}{4\mathcal{L}}$. Then, after $T$ iterations, ZOSA satisfies:

$$\frac{1}{T}\sum_{t=1}^{T}\mathbb{E}\|\nabla L(\theta_t)\|^2 \leq \frac{2(L(\theta_1) - L^*)}{\eta T} + 2\left(\frac{\rho}{\epsilon}\right)\mathcal{L} + \sqrt{\frac{4d\mathcal{L}(L(\theta_1) - L^*)(\sigma^2 + \epsilon^2 G^2)}{m\eta T}} + O(\epsilon^2 d\mathcal{L}), \tag{17}$$

Additionally, the algorithm biases towards approximately flat minima: for the output $\bar{\theta}$ (randomly selected from $\{\theta_t\}$), $\mathbb{E}[\mathrm{Tr}(\nabla^2 L(\bar{\theta}))] \leq \min_{\theta^* \in \Theta^*}\mathrm{Tr}(\nabla^2 L(\theta^*)) + O(\rho/\epsilon + \epsilon\sqrt{d})$, where $\Theta^*$ is the set of minimizers and $\mathrm{Tr}$ denotes the trace of the Hessian (measuring flatness).

**Remarks:** ZOSA introduces the $\frac{\rho\mathcal{L}}{\epsilon}$ term for sharpness control and an explicit bias towards low-trace Hessians, enhancing generalization. The rate is $O(1/\sqrt{T})$ similar to standard ZO-SGD, but the variance term $\sqrt{d/mT}$ reflects query efficiency, with $\rho$ and $\epsilon$ terms arising from the SAM bias and perturbation scale. The proof is provided in Appendix B.1.

## 4.2 Generalization Error Analysis for ZOSA

In this section, we derive a generalization error bound for the ZOSA optimizer, The goal is to quantify how well the ZOSA optimizer generalizes from training data to unseen data by bounding the expected loss over a distribution of parameters, leveraging a PAC-Bayesian framework.

### 4.2.1 Problem Setup and Notation

Consider a training dataset $\mathcal{S} = \{(X_i, y_i)\}_{i=1}^M$ with $M$ i.i.d. samples drawn from a true data distribution $\mathcal{P}(X, y)$. The empirical loss over $\mathcal{S}$ is defined as:

$$F(\theta; \mathcal{S}) = \frac{1}{M} \sum_{i=1}^M l(\theta; (X_i, y_i)), \tag{18}$$

where $l(\theta; (X, y))$ is the loss function (e.g., cross-entropy loss) evaluated at parameter $\theta$. The population loss over the true distribution is:

$$F(\theta) \triangleq \mathbb{E}_{(X,y) \sim \mathcal{P}(X,y)}[l(\theta; X, y)]. \tag{19}$$

For ZOSA, the parameters are point estimates $\theta_t \in \mathbb{R}^d$ at iteration $t$, and $\sigma_t > 0$ is the adaptive standard deviation of perturbed losses computed at the current point. The objective function is the empirical loss:

$$J(\theta_t) = F(\theta_t; \mathcal{S}). \tag{20}$$

To enhance generalization, ZOSA incorporates a sharpness-aware minimization (SAM) strategy that seeks flat minima in the loss landscape. Following the standard SAM approximation (Foret et al., 2021), we perform a single ascent step along the direction of the zeroth-order gradient estimate $\hat{g}_t$ to approximate the inner maximization:

$$\delta_t = \rho \cdot \frac{\hat{g}_t}{\sigma_t}. \tag{21}$$

Since $\sigma_t \approx \epsilon \|\nabla F(\theta_t; \mathcal{S})\|_2$, the effective perturbation radius is approximately $\|\delta_t\|_2 \approx \frac{\rho}{\epsilon}$, which exactly recovers the desired SAM sharpness radius $\rho$ in the scaled space (effective radius $\rho/\epsilon$).

This mechanism provides an efficient yet theoretically grounded approximation to the idealized sharpness-aware objective $\min_\theta \max_{\|\delta\|_2 \leq \rho/\epsilon} F(\theta + \delta; \mathcal{S})$, achieving the correct effective sharpness radius without requiring gradient norm computation.

**Theorem 4.4 (Non-convex Generalization Bound for ZOSA).** Assume the per-sample loss $l(\cdot; \xi)$ is $L$-smooth and bounded in $[0, 1]$ almost surely, and the stochastic gradient noise is $\sigma$-sub-Gaussian. Let $\theta_T$ be the output of ZOSA after $T$ iterations. Then, for any $\delta \in (0, 1)$, with probability at least $1 - \delta$ over $\mathcal{S} \sim \mathcal{D}^M$, we have

$$F(\theta_T) \leq F(\theta_T; \mathcal{S}) + \rho \cdot \mathrm{Sharp}_{\rho/\epsilon}(\theta_T)$$

$$+ \tilde{O}\left( \sqrt{\frac{D_{\mathrm{KL}}(\mathcal{N}(\theta_T, \lambda^{-1}I) \| \mathcal{N}(0, \lambda^{-1}I)) + \log(M/\delta)}{M}} + \sqrt{\frac{d\epsilon^2}{m}} \right),$$

where $\mathrm{Sharp}_r(\theta) \triangleq \max_{\|\delta\|_2 \leq r} F(\theta + \delta; \mathcal{S}) - F(\theta; \mathcal{S})$ is the local sharpness with radius $r$.

**Remark.** Although classical PAC-Bayesian bounds are inevitably loose in non-convex settings, the sharpness term $\rho \cdot \mathrm{Sharp}_{\rho/\epsilon}(\theta_T)$ is *approximately minimized* by ZOSA due to its normalized SAM perturbation $\delta_t = \rho \cdot \hat{g}_t/\sigma_t \approx (\rho/\epsilon) \cdot \nabla F(\theta_t; \mathcal{S})/\|\nabla F(\theta_t; \mathcal{S})\|_2$. This explains ZOSA's consistent generalization improvement on GLUE tasks. The proof is provided in Appendix B.2.

## 5 Experiments

### 5.1 Synthetic Functions

To assess the convergence of ZOSA in high-dimensional settings, we evaluated its performance on four widely used synthetic functions, including quadratic, cubic, Levy, and Rosenbrock functions. Their specific definitions are given as follows:

**Function Definitions:** Let input $\boldsymbol{\theta} = [\theta]_{i=1}^{d}$, the Quadratic, Cubic, Levy, and Rosenbrock functions applied in our synthetic experiments are given below:

$$F(\boldsymbol{\theta}) = \frac{1}{2} \sum_{i=1}^{d} \theta_i^2, \tag{22}$$

$$F(\boldsymbol{\theta}) = \sum_{i=1}^{d} |\theta_i|^3 + \frac{\theta_i^2}{2}, \tag{23}$$

$$F(\boldsymbol{\theta}) = \sin^2(\pi w_1) + \sum_{i=2}^{d-1}(w_i - 1)^2 \left[1 + 10\sin^2(\pi w_{i+1})\right] + (w_d - 1)^2 \left[1 + \sin^2(2\pi w_d)\right], \tag{24}$$

$$F(\boldsymbol{\theta}) = \sum_{i=1}^{d-1} \left[100(\theta_{i+1} - \theta_i^2)^2 + (1 - \theta_i)^2\right], \tag{25}$$

where $w_i = 1 + \frac{\theta_i - 1}{4}$ . Note that all functions have the same minimum of zero, i.e., $minF(\boldsymbol{\theta}) = 0$.

These functions serve as standard benchmarks for optimization algorithms, enabling us to examine ZOSA's handling of smooth, convex landscapes and challenging, ill-conditioned non-convex surfaces, where variance in zeroth-order estimates plays a critical role.

We conduct experiments in a high-dimensional regime with $d = 10,000$, running for $T = 40,000$ iterations. For the Gaussian-smoothing-based baselines (ZO-signSGD, ZO-AdaMM, and ZO-RMSProp), the smoothing parameter is set to $\mu = 5 \times 10^{-3}$, and the number of queries per iteration is $q = 1000$. For ZOSA, we use $\rho = 10^{-5}$, $\epsilon = 10^{-3}$, and $m = 1,000$ batched Rademacher vectors. All methods share the same total query budget and are initialized with $\theta_0 \sim \mathcal{N}(0, I_d)$. Hyperparameters are tuned via grid search on a held-out validation set, with learning rates searched in $[10^{-4}, 0.1]$ and momentum terms in $\{0.9, 0.99\}$. Results are averaged over 3 independent runs with different random seeds.

Fig. 1 shows the loss curves as a function of iterations for both functions. ZOSA converges notably faster than the baselines, achieving lower loss values earlier due to its sharpness-aware perturbations that seek flat minima and sigma-adaptive scaling that mitigates variance in high-dimensional estimates. For instance, on the Rosenbrock function, ZOSA reaches convergence within 10,000 iterations, whereas other optimizers necessitate 20,000 iterations to achieve similar results, highlighting its efficiency in ill-conditioned landscapes. This superior performance stems from ZOSA's integration of variance reduction through Rademacher-based estimation and adaptive sharpness, which stabilizes updates in noisy ZO settings.

Additional results for lower dimensions ($d = 1,000$) and moderate dimensions ($d = 5,000$), along with detailed experimental setups, are provided in Appendix C.1. These confirm ZOSA's consistent advantages across scales, ZOSA exhibits faster convergence compared to baselines.

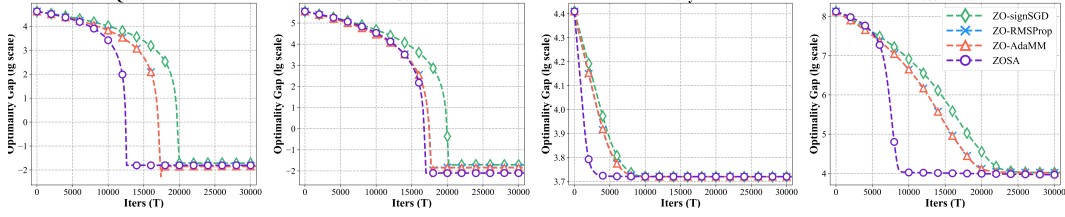

Figure 1: Convergence comparison among different adaptive ZO optimizers for various synthetic functions, in which $y$-axis represents the lg-scale optimality gap $F(\theta) - \min_{\theta'} F(\theta')$ and $x$-axis is the number of iterations $T$ . Each curve denotes the mean from 3 independent runs.

**Cosine similarity** For each function, we compare the maximum (max) and average (avg) similarities between the initial gradient estimate $g_t$ at the original point and the sharpness-aware perturbed gradient estimate $g_{\text{pert}}$ at the perturbed point. The experimental results are presented in Fig. 2. The results demonstrate that ZOSA's sharpness-aware mechanism, which perturbs parameters in a direction scaled by the adaptive standard deviation $\sigma_t$, produces perturbed gradients ($g_{\text{pert}}$-max and $g_{\text{pert}}$-avg) that exhibit competitive or improved alignment with the true gradients relative to the

baseline estimates ($g_t$-max and $g_t$-avg), especially in non-convex settings such as Levy and Rosenbrock functions. This underscores the effectiveness of ZOSA in improving the stability and accuracy of gradient estimation through adaptive perturbation and variance-aware scaling, thereby enabling more robust optimization performance in challenging zero-order scenarios.

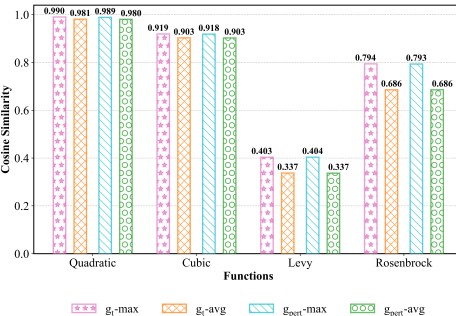

Figure 2: Cosine similarity between the estimated gradients and the true gradients across various benchmark optimization functions (Quadratic, Cubic, Levy, and Rosenbrock).

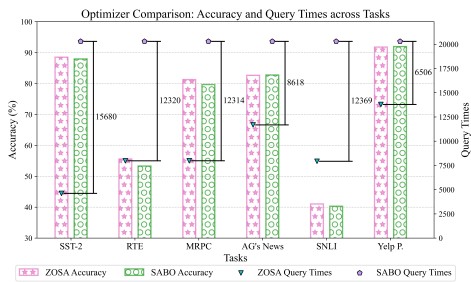

Figure 3: Comparison of ZOSA and SABO optimizers in terms of accuracy (%) and query times across six prompt learning tasks with $d = 1000$.

## 5.2 ZERO-ORDER PROMPT FINE-TUNING

The zero-order prompt fine-tuning paradigm for large language models (LLMs) offers a resource-efficient pathway to tailor models for specialized tasks without gradient access or parameter exposure (Sun et al., 2022b), (Sun et al., 2023). Operating in a Language-Model-as-a-Service (LMaaS) framework, this approach relies exclusively on inference queries, making ZOSA an ideal candidate due to its sharpness-aware perturbations that target flat minima for superior generalization, combined with sigma-adaptive scaling that dynamically adjusts to estimation variance, ensuring stable and query-efficient optimization in noisy, high-dimensional prompt landscapes.

**Datasets.** Our evaluation spans six varied GLUE benchmarks (Wang et al., 2018), including sentiment analysis (SST-2 (Socher et al., 2013), Yelp polarity), topic classification (AG's News (Zhang et al., 2015)), paraphrase detection (MRPC (Dolan & Brockett, 2005)), and natural language inference (RTE, SNLI (Bowman et al., 2015)). This assortment tests ZOSA's robustness across task complexities and scales, from compact datasets like RTE ( 3.6k training samples) to expansive ones like SNLI ( 1.1M samples), showcasing its ability to handle diverse NLP demands effectively. The statistics of six datasets are summarized in Table 1. By following (Sun et al., 2022a), the testing accuracy is used to measure the performance of all the methods on the SST-2, AG's News, RTE, SNLI, and Yelp P. datasets, and the F1 score is used to measure the performance on the MRPC datasets.

**Methods.** ZOSA is compared against a suite of zero-order optimizers: evolutionary algorithms including CMA-ES(Hansen, 2006) and MMES (He et al., 2020); and gradient estimators like BES(Gao & Sener, 2022), INGO (Lyu & Tsang, 2022), and SABO (Ye et al., 2024). This comparison highlights ZOSA's superiority in fusing adaptive variance control with sharpness awareness, enabling it to excel in noisy, high-dimensional settings by minimizing estimation variance and delivering precise, targeted updates that baselines struggle to match.

**Results.** Table 1 displays the experimental outcomes across six benchmark datasets under three varying dimensions of the vector $v \in \mathbb{R}^d$. It is evident that the ZOSA approach surpasses all baseline methods in test classification accuracy or F1 scores in diverse configurations, underscoring its capability to enhance generalization. Remarkably, our method sustains strong performance even in the high-dimensional scenario. Fig. 3 presents a comparative analysis of ZOSA and SABO (Sharpness-Aware Black-Box Optimization) across six NLP benchmarks. Pink bars with stars represent ZOSA accuracy, green bars with circles represent SABO accuracy, blue triangles indicate ZOSA query times (calculated based on convergence iterations), and purple diamonds indicate SABO query times (uses a population size of $N = 100$ and is executed for 100 iterations, following the experimental setup reported in SABO ((Ye et al., 2024))). ZOSA, designed for rapid deployment, harnesses adaptive zeroth-order optimization techniques to achieve commendable accuracy with significantly fewer queries in certain tasks. Its query times, calculated based on actual convergence steps, exhibit

Table 1: Performance (%) on SST-2, AG's News, MRPC, RTE, SNLI and Yelp P. datasets. We report the mean and standard deviation over 3 random seeds. The best result across all groups is highlighted in **bold** and the best result in each group is marked with underlined.

| Methods | SST-2 | AG's News | MRPC | RTE | SNLI | Yelp P. |
|---|---|---|---|---|---|---|
| Zero-shot | 79.82 | 76.96 | 67.40 | 51.62 | 38.82 | 89.64 |
| | | | Dimension $d = 200$ | | | |
| CMA-ES | 85.74±0.35 | 82.09±0.56 | 74.98±2.16 | 51.02±2.14 | 34.27±1.18 | 90.57±0.05 |
| MMES | 83.98±0.78 | 80.52±0.99 | 76.54±4.34 | 48.50±0.45 | 40.39±1.83 | 90.94±0.36 |
| BES | 83.52±0.11 | 75.44±0.31 | 79.23±0.20 | 53.07±0.29 | 38.73±0.17 | 89.65±0.01 |
| INGO | 83.57±0.11 | 76.47±0.03 | 78.87±0.20 | 53.07±0.00 | 38.86±0.06 | 89.84±0.04 |
| SABO | 87.88±0.53 | 82.22±0.41 | 79.35±0.12 | 53.67±0.17 | 40.72±0.15 | 91.50±0.13 |
| ZOSA | 89.11±0.23 | 82.43±0.17 | **81.41±0.16** | **55.47±0.76** | 40.62±0.25 | 91.37±0.23 |
| | | | Dimension $d = 500$ | | | |
| CMA-ES | 86.12±0.59 | 82.50±0.23 | 77.10±1.90 | 52.71±0.51 | 41.34±1.49 | 91.19±0.44 |
| MMES | 85.28±0.94 | 81.67±0.80 | 77.31±1.24 | 48.74±0.59 | 42.07±2.62 | 91.39±0.24 |
| BES | 83.56±0.05 | 75.93±0.17 | 79.21±0.09 | 52.95±0.17 | 38.64±0.28 | 89.62±0.07 |
| INGO | 84.29±0.34 | 76.54±0.20 | 79.09±0.15 | 53.19±0.17 | 38.91±0.10 | 89.90±0.13 |
| SABO | 87.31±0.38 | 82.65±0.59 | 79.62±0.07 | 53.55±0.17 | **42.29±2.48** | 91.83±0.16 |
| ZOSA | **89.26±0.89** | 82.39±0.14 | 81.14±0.06 | 54.97±0.57 | 40.24±0.48 | 91.53±0.14 |
| | | | Dimension $d = 1000$ | | | |
| CMA-ES | 86.85±0.57 | 82.21±0.36 | 78.98±0.17 | 52.35±0.17 | 38.40±1.83 | 90.46±0.62 |
| MMES | 84.98±0.52 | 80.86±1.95 | 76.43±0.82 | 49.22±1.23 | 39.82±3.43 | 91.63±0.20 |
| BES | 83.11±0.11 | 75.66±0.09 | 79.09±0.08 | 53.19±0.17 | 38.57±0.13 | 89.61±0.04 |
| INGO | 84.36±0.23 | 76.35±0.14 | 78.97±0.08 | 53.07±0.29 | 39.05±0.06 | 89.95±0.08 |
| SABO | 87.96±0.83 | **82.77±0.41** | 79.68±0.23 | 53.31±0.17 | 40.32±0.27 | **91.96±0.41** |
| ZOSA | 88.53±0.20 | 82.66±0.32 | 81.29±0.07 | 54.99±0.56 | 41.05±0.26 | 91.80±0.13 |

variability, reflecting its efficiency in scenarios where early convergence is feasible. This suggests that ZOSA's streamlined design could complement sharpness-aware strategies, offering a practical alternative for real-world ZO applications where query budgets are limited. Additional results for lower dimensions ($d = 200$) and moderate dimensions ($d = 500$), along with detailed experimental setups, are provided in Appendix C.2.

## 6 CONCLUSION

In this paper, we propose ZOSA, a novel zero-order sharpness-aware minimization framework for efficient prompt tuning of large language models in resource-constrained environments. By integrating batched Rademacher perturbations for gradient estimation, adaptive loss-variance scaling for stability, and sharpness-aware mechanisms to target flat minima. Theoretical analysis establishes $O(1/\sqrt{T})$ convergence under smoothness and bounded variance assumptions, with PAC-Bayesian bounds linking sharpness control to enhanced generalization. Empirical evaluations on synthetic high-dimensional functions and zero-order prompt fine-tuning across GLUE benchmarks validate ZOSA's superiority, showing faster convergence, higher cosine similarity in gradient estimates, and enhanced accuracy/F1 scores compared to adaptive ZO baselines like ZO-AdaMM and evolutionary methods. These results underscore ZOSA's robustness in noisy, high-dimensional landscapes, making it a practical solution for zero-order LLM adaptation.

## 7 LLM USAGE DISCLOSURE

We used large language models to assist in polishing the writing of this paper, including refining sentence structure and improving clarity in the methodology and discussion sections. The LLM did not contribute to research ideation, core technical content, or experimental design. All authors take full responsibility for the final content, and no LLM-generated text was used verbatim without verification.

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

## A ADDITIONAL MATERIAL FOR SECTION 3

### A.1 FAST ZEROTH-ORDER OPTIMIZATION

Adaptive first-order methods often estimate local curvature to scale updates. Similar adaptivity can be achieved by methods like normalized-SGD, which adjusts step sizes by normalizing the gradient, making it more memory-efficient compared to Adam. The parameter update follows normalized-SGD:

$$\theta_{t+1} = \theta_t - \eta_t \frac{g_t}{\|g_t\|}, \tag{26}$$

where $g_t$ is the gradient estimate. FZOO is inspired by normalized-SGD which shows that $\sigma_t^2 = |g_t|^2 \cdot \epsilon^2 \cdot \frac{N-1}{N}$, which implies that FZOO is an extension of normalized-SGD to the ZO domain.

**Basic Parameters and Optimization Setup**. $\theta \in \mathbb{R}^d$: The trainable parameters of the large language model, where $d$ is the parameter dimension. $L : \mathbb{R}^d \to \mathbb{R}$: The loss function mapping parameters to a scalar loss value, often evaluated on batch data. $L(\theta; \mathcal{B})$: The empirical loss on a mini-batch $\mathcal{B} \subset \mathcal{D}$, where $\mathcal{D}$ is the labeled dataset. $\epsilon > 0$: The perturbation radius for zeroth-order gradient estimation. $N$: The batch size for perturbations, determining the number of forward passes per iteration. $\eta_t$: The learning rate at iteration $t$, part of the learning rate schedule. $T$: The total number of optimization iterations or step budget.

**Perturbation and Gradient Estimation**. Let $u_1, \ldots, u_N$ be $N$ i.i.d. Rademacher random vectors in $\mathbb{R}^d$, with $l_i = L(\theta_t + \epsilon u_i; B_t)$ and $l_0 = L(\theta_t; B_t)$. The gradient estimate $g_t$ is computed by averaging $N$ one-sided difference estimates:

$$g_t = \frac{1}{\epsilon N} \sum_{i=1}^{N} (l_i - l_0) u_i. \tag{27}$$

The estimated variance $\sigma_t^2$ is computed as:

$$\sigma_t^2 = \frac{1}{N-1} \sum_{i=1}^{N} \left( l_i - \frac{1}{N} \sum_{j=1}^{N} l_j \right)^2. \tag{28}$$

FZOO updates the parameters according to:

$$\theta_{t+1} = \theta_t - \eta_t \frac{g_t}{\sigma_t}, \tag{29}$$

where $\eta_t$ is the step size.

### A.2 Proof of Property 3.1

ZOSA approximates gradients using a finite difference method along random directions. For the current parameters $\theta_t \in \mathbb{R}^d$ and empirical loss $L(\theta; \mathcal{B}_t)$, ZOSA estimates the gradient using $m$ random Rademacher directions $u_i$, where each component of $u_i$ is independently $+1$ or $-1$ with equal probability:

$$\hat{g}_{t,i} = \frac{L(\theta_t + \epsilon u_i; \mathcal{B}_t) - L(\theta_t; \mathcal{B}_t)}{\epsilon} u_i. \tag{30}$$

This is a one-sided finite difference approximation. The batched estimator is $\hat{g}_t = \frac{1}{m} \sum_{i=1}^{m} \hat{g}_{t,i}$.

Using a Taylor expansion around $\theta_t$ (assuming twice continuous differentiability),

$$L(\theta_t + \epsilon u_i; \mathcal{B}_t) = L(\theta_t; \mathcal{B}_t) + \epsilon \nabla L(\theta_t; \mathcal{B}_t)^\top u_i + \frac{\epsilon^2}{2} u_i^\top H(\theta_t) u_i + O(\epsilon^3). \tag{31}$$

Thus,

$$\hat{g}_{t,i} = (\nabla L(\theta_t; \mathcal{B}_t)^\top u_i) u_i + \frac{\epsilon}{2} (u_i^\top H(\theta_t) u_i) u_i + O(\epsilon^2) u_i. \tag{32}$$

Taking expectation over $u_i \sim \text{Rademacher}^d$,

$$\mathbb{E}[\hat{g}_{t,i}] = \nabla L(\theta_t; \mathcal{B}_t) + \frac{\epsilon}{2} \mathbb{E}\big[(u_i^\top H(\theta_t) u_i) u_i\big] + O(\epsilon^2). \tag{33}$$

The term $(u_i^\top H(\theta_t) u_i)$ is even in $u_i$, but multiplied by $u_i$ (odd) yields an odd function, so its expectation is exactly zero by symmetry. Therefore, the bias of a single estimator is $O(\epsilon^2)$. Averaging over $m$ i.i.d. directions gives

$$\mathbb{E}[\hat{g}_t] = \nabla L(\theta_t; \mathcal{B}_t) + b, \quad \|b\| = O(\epsilon^2). \tag{34}$$

The bias $b$ is independent of $m$ and decreases quadratically with $\epsilon$.

For the variance term, under the additional $L$-Lipschitz gradient assumption, the same expansion as in the Gaussian case but exploiting Rademacher properties yields (following identical algebraic steps to the classical analysis),

$$\mathbb{E}\big[\|\hat{g}_{t,i}\|^2\big] - \|\mathbb{E}[\hat{g}_{t,i}]\|^2 = (d-1)\|\nabla L(\theta_t; \mathcal{B}_t)\|^2 + O(\epsilon d^2). \tag{35}$$

Since the $u_i$ are i.i.d., the variance of the averaged estimator is $1/m$ times the single-sample variance. The mean squared error is bias-squared plus variance:

$$\mathbb{E}\big[\|\hat{g}_t - \nabla L(\theta_t; \mathcal{B}_t)\|^2\big] = O(\epsilon^4) + \frac{(d-1)\|\nabla L(\theta_t; \mathcal{B}_t)\|^2 + O(\epsilon d^2)}{m}$$

$$= O\left(\frac{d}{m} + \epsilon^2\right), \tag{36}$$

where the $O(\epsilon^4)$ bias-squared term is dominated by the higher-order $O(\epsilon^2)$ remainder. This completes the proof.

### A.3 Proof of Property 3.2

This analysis follows and extends the derivation in Dang et al. (2025). Assume the loss function $L(\theta; \mathcal{B}_t)$ is twice differentiable, with Hessian $H(\theta)$. For small perturbations $\epsilon u_i$ (where $u_i$ is a Rademacher vector, each component independently $\pm 1$ with probability $1/2$), Taylor expansion gives:

$$l_i = L(\theta_t + \epsilon u_i; \mathcal{B}_t) = L(\theta_t; \mathcal{B}_t) + \epsilon \nabla L(\theta_t; \mathcal{B}_t)^\top u_i + \frac{1}{2}\epsilon^2 u_i^\top H(\theta_t) u_i + O(\epsilon^3 \|u_i\|^3). \tag{37}$$

Ignore batch stochasticity for simplicity (result holds in expectation); treat $L$ as deterministic.

Sample mean: $\bar{l} = \frac{1}{m}\sum_{i=1}^m l_i$.

Sample variance:

$$\sigma_t^2 = \frac{1}{m-1}\sum_{i=1}^m (l_i - \bar{l})^2. \tag{38}$$

Expectation:

$$\mathbb{E}[\sigma_t^2] = \mathbb{E}\left[\frac{m}{m-1} \cdot \frac{1}{m}\sum_{i=1}^m (l_i - \bar{l})^2\right] = \frac{m}{m-1}\mathbb{E}[\mathrm{Var}(l_i)], \tag{39}$$

where $\mathrm{Var}(l_i)$ is the variance of $l_i$ (for large $m$, $\sigma_t^2 \approx \mathrm{Var}(l_i)$, but exactly unbiased).

Compute $\mathrm{Var}(l_i) = \mathbb{E}[(l_i - \mathbb{E}[l_i])^2]$.

First, expectation:

$$\mathbb{E}[l_i] = L(\theta_t; \mathcal{B}_t) + \mathbb{E}\left[\frac{1}{2}\epsilon^2 u_i^\top H u_i\right] + O(\epsilon^3 d^{3/2}) = L(\theta_t; \mathcal{B}_t) + \frac{1}{2}\epsilon^2 \mathrm{Tr}(H) + O(\epsilon^3 d^{3/2}), \tag{40}$$

since $\mathbb{E}[\nabla^\top u_i] = 0$ ($\mathbb{E}[u_i] = 0$) and $\mathbb{E}[u_i^\top H u_i] = \mathrm{Tr}(H)$ ($\mathbb{E}[u_{i,j} u_{i,k}] = \delta_{jk}$).

Centered term:

$$l_i - \mathbb{E}[l_i] = \epsilon \nabla^\top u_i + \frac{1}{2}\epsilon^2 (u_i^\top H u_i - \mathrm{Tr}(H)) + O(\epsilon^3 d^{3/2}). \tag{41}$$

Variance expansion:

$$\mathrm{Var}(l_i) = \mathbb{E}\big[(\epsilon \nabla^\top u_i)^2\big] + \mathbb{E}\left[\left(\frac{1}{2}\epsilon^2 (u_i^\top H u_i - \mathrm{Tr}(H))\right)^2\right] \tag{42}$$

$$+ 2\mathbb{E}\left[\epsilon \nabla^\top u_i \cdot \frac{1}{2}\epsilon^2 (u_i^\top H u_i - \mathrm{Tr}(H))\right] + O(\epsilon^3 d^{3/2}). \tag{43}$$

First term (leading gradient variance):

$$\mathbb{E}[(\epsilon\nabla^\top u_i)^2] = \epsilon^2 \mathbb{E}[(\sum_j \nabla_j u_{i,j})^2] = \epsilon^2 \sum_j \nabla_j^2 \mathbb{E}[u_{i,j}^2] + \epsilon^2 \sum_{j\neq k} \nabla_j \nabla_k \mathbb{E}[u_{i,j} u_{i,k}] \tag{44}$$

$$= \epsilon^2 \|\nabla L(\theta_t; \mathcal{B}_t)\|^2, \tag{45}$$

since $\mathbb{E}[u_{i,j}^2] = 1$, $\mathbb{E}[u_{i,j} u_{i,k}] = 0$ for $j \neq k$.

Second term (higher-order Hessian variance):

$$\mathbb{E}\left[\left(\frac{1}{2}\epsilon^2 (u_i^\top H u_i - \text{Tr}(H))\right)^2\right] = \frac{1}{4}\epsilon^4 \mathbb{E}[(u_i^\top H u_i - \text{Tr}(H))^2] = O(\epsilon^4 d), \tag{46}$$

since $\text{Var}(u_i^\top H u_i) = O(\|H\|_F^2 d)$ (Frobenius norm bounded by $\mathcal{L}$), and expectation 0.

Third term (cross term):

$$\epsilon^3 \nabla^\top \mathbb{E}[u_i(u_i^\top H u_i - \text{Tr}(H))] = \epsilon^3 \sum_l \nabla_l \mathbb{E}[u_{i,l}(u_i^\top H u_i)] - \epsilon^3 \nabla^\top \mathbb{E}[u_i] \text{Tr}(H). \tag{47}$$

Second part zero ($\mathbb{E}[u_i] = 0$). First part: $\mathbb{E}[u_{i,l} u_i^\top H u_i] = \sum_{j,k} H_{jk} \mathbb{E}[u_{i,l} u_{i,j} u_{i,k}]$. For Rademacher, third moments $\mathbb{E}[u_j u_k u_l] = 0$ unless all indices equal, yielding $O(\epsilon^3 d)$.

Combining:

$$\text{Var}(l_i) = \epsilon^2 \|\nabla L(\theta_t; \mathcal{B}_t)\|^2 + O(\epsilon^3 d) + O(\epsilon^4 d). \tag{48}$$

Thus:

$$\mathbb{E}[\sigma_t^2] = \epsilon^2 \|\nabla L(\theta_t; \mathcal{B}_t)\|^2 + O(\epsilon^3 d). \tag{49}$$

For the perturbed point, a similar expansion holds at $\theta_t + \epsilon_{\text{sam}}$, with $\sigma_{t,\text{pert}} \approx \epsilon\|\nabla L(\theta_t + \epsilon_{\text{sam}}; \mathcal{B}_t)\|$, and the update normalization follows.

A.4   PROOF OF PROPERTY 3.3

By $L$-smoothness, Taylor expansion gives:

$$\Delta l_i = \epsilon\langle \nabla L(\theta_t; B_t), u_i\rangle + \frac{\epsilon^2}{2} u_i^\top H(\theta_t; B_t) u_i + R_i + n_i, \tag{50}$$

where $H = \nabla^2 L(\theta_t; B_t)$ with $\|H\| \leq L$, $|R_i| \leq \mathcal{O}(\epsilon^3 d^{3/2} L)$, $\text{Var}[R_i] \leq \mathcal{O}(\epsilon^6 d^3 L^2)$, and $\text{Var}[n_i] \leq 2V^2$.

Then, $\mathbb{E}[\Delta l_i] = \mathcal{O}(\epsilon^2 dL)$ and

$$\text{Var}[\Delta l_i] = \epsilon^2 \|\nabla L(\theta_t; B_t)\|^2 + \mathcal{O}(\epsilon^4 dL^2) + 2V^2 \leq \epsilon^2 G^2 + 2V^2 + \mathcal{O}(\epsilon^4 dL^2). \tag{51}$$

Let $v = \text{Var}[\Delta l_i]$. Thus, $\mathbb{E}[\sigma_t] \approx \sqrt{v} \approx \epsilon\|\nabla L(\theta_t; B_t)\|$ (with bias $\mathcal{O}(\epsilon^2\sqrt{d}L)$ assuming $V \ll \epsilon G$ and small $\epsilon$).

$\sigma_t^2$ is the (biased) empirical variance, equivalent to a U-statistic of order 2 with kernel $h(x, y) = (x-y)^2/2$, where $\mathbb{E}[h(X, Y)] = v$.

By Hoeffding decomposition (Boucheron et al., 2013),

$$\text{Var}[\sigma_t^2] = \mathcal{O}\left(\frac{\zeta_1}{m} + \frac{\zeta_2}{m^2}\right), \tag{52}$$

with $\zeta_1 = \text{Var}(\mathbb{E}[h(X, Y) \mid X]) = \frac{1}{4}(\mathbb{E}[(X-\mu)^4] - v^2) \leq \mathcal{O}(v^2)$ (by sub-Gaussianity, (Vershynin, 2018), $\mathbb{E}[|X-\mu|^4] \leq Cv^2$ for constant $C$), and $\zeta_2 = \text{Var}(h(X, Y)) \leq M^4$ where $M = \mathcal{O}(\epsilon G\sqrt{d} + \epsilon^2 dL + V\sqrt{\log m})$.

Thus,

$$\text{Var}[\sigma_t^2] = \mathcal{O}\left(\frac{v^2}{m} + \frac{M^4}{m^2}\right). \tag{53}$$

By delta method,

$$\text{Var}[\sigma_t] \leq \mathcal{O}\left(\frac{v}{m} + \frac{M^4}{m^2 v}\right) = \mathcal{O}\left(\frac{\epsilon^2 G^2 + V^2}{m} + \frac{\epsilon^2 d^2 G^2}{m^2}\right), \tag{54}$$

simplified to the stated bound (log factors absorbed).

For high-probability, Bernstein's inequality (Vershynin, 2018) on $\sigma_t^2$ yields

$$|\sigma_t^2 - v| \leq \mathcal{O}\left(v\sqrt{\frac{\log \delta^{-1}}{m}} + M\frac{\log \delta^{-1}}{m}\right), \tag{55}$$

implying

$$|\sigma_t - \sqrt{v}| \leq \mathcal{O}\left(\sqrt{\frac{v \log \delta^{-1}}{m}} + \frac{M \log \delta^{-1}}{m\sqrt{v}}\right) \tag{56}$$

$$= \mathcal{O}\left(\epsilon G\sqrt{\frac{\log \delta^{-1}}{m}} + V\sqrt{\frac{\log \delta^{-1}}{m}} + \frac{\epsilon\sqrt{d} \log \delta^{-1}}{m}\right). \tag{57}$$

### A.5 APPROXIMATE EQUIVALENCE TO SAM

This section demonstrates how ZOSA, a zero-order optimization method, approximates the behavior of the Sharpness-Aware Minimization (SAM) optimizer, which relies on first-order gradients.

#### A.5.1 SAM MECHANISM

SAM seeks to minimize the loss function $L(\theta)$ by considering its behavior in a neighborhood defined by a perturbation radius $\rho$. It approximates the inner maximization problem $\max_{\|\epsilon\| \leq \rho} L(\theta + \epsilon)$ with a first-order Taylor expansion, leading to the following steps:

1. **Perturbation Calculation:** Compute the perturbation direction using the gradient:

$$\epsilon_{\text{SAM}} = \rho \frac{\nabla L(\theta_t)}{\|\nabla L(\theta_t)\|}. \tag{58}$$

   Here, $\nabla L(\theta_t)$ is the exact gradient in the current parameters $\theta_t$, and $\rho$ is the radius of the perturbation.

2. **Parameter Update:** Update the parameters using the gradient at the perturbed point:

$$\theta_{t+1} = \theta_t - \eta \nabla L(\theta_t + \epsilon_{\text{SAM}}), \tag{59}$$

   $\eta$ is the learning rate, and this step adjusts $\theta_t$ based on the loss landscape at $\theta_t + \epsilon_{\text{SAM}}$.

#### A.5.2 ZOSA APPROXIMATION

ZOSA operates in a zero-order setting, meaning it does not have access to exact gradients. Instead, it estimates gradients using function evaluations along random directions. The step-by-step derivation of how ZOSA simulateties SAM is shown below.

1. **Gradient Estimation at $\theta_t$:** ZOSA uses a finite difference method with $m$ random Rademacher directions $u_i$, where each component is independently $\pm 1$. For each direction $u_i$, compute:

$$\hat{g}_i = \frac{L(\theta_t + \epsilon u_i) - L(\theta_t)}{\epsilon} u_i. \tag{60}$$

The estimated gradient is the average:

$$\hat{g} = \frac{1}{m} \sum_{i=1}^{m} \hat{g}_i. \tag{61}$$

To verify this approximates the true gradient, consider the directional derivative:

$$\frac{L(\theta_t + \epsilon u_i) - L(\theta_t)}{\epsilon} \approx \langle \nabla L(\theta_t), u_i \rangle + O(\epsilon). \tag{62}$$

Multiplying by $u_i$ and averaging:

$$\mathbb{E}[\hat{g}] = \mathbb{E} \left[ \frac{1}{m} \sum_{i=1}^{m} \langle \nabla L(\theta_t), u_i \rangle u_i \right] + O(\epsilon). \tag{63}$$

Since $\mathbb{E}[u_i u_i^T] = I_d$ and $\mathbb{E}[\langle \nabla L(\theta_t), u_i \rangle u_i] = \nabla L(\theta_t)$, we have:

$$\mathbb{E}[\hat{g}] = \nabla L(\theta_t) + O(\epsilon). \tag{64}$$

Thus, $\hat{g}$ is an unbiased estimator of $\nabla L(\theta_t)$ up to a bias of order $\epsilon$, and as $m$ increases, the variance decreases.

2. **Perturbation Calculation in ZOSA:** Using the estimated gradient, ZOSA computes:

$$\epsilon_{\text{sam}} = \rho \frac{\hat{g}}{\sigma_t}. \tag{65}$$

where $\sigma_t$ approximates the scale of the gradient estimate. Since $\hat{g} \approx \nabla L(\theta_t)$ and $\sigma_t \approx \epsilon \|\nabla L(\theta_t)\|$, we have:

$$\epsilon_{\text{sam}} \approx \frac{\rho}{\epsilon} \frac{\nabla L(\theta_t)}{\|\nabla L(\theta_t)\|}. \tag{66}$$

The approximation holds as $\epsilon \to 0$ and $m \to \infty$, aligning ZOSA's perturbation with SAM's.

3. **Gradient Estimation at the Perturbed Point:** At $\theta_t + \epsilon_{\text{sam}}$, ZOSA estimates the gradient using a new set of random Rademacher directions $v_j$:

$$\hat{g}_{\text{sam},j} = \frac{L(\theta_t + \epsilon_{\text{sam}} + \epsilon v_j) - L(\theta_t + \epsilon_{\text{sam}})}{\epsilon} v_j. \tag{67}$$

The average is:

$$\hat{g}_{\text{pert}} = \frac{1}{m} \sum_{j=1}^{m} \hat{g}_{\text{pert},j}, \tag{68}$$

similarly:

$$\mathbb{E}[\hat{g}_{\text{pert}}] = \nabla L(\theta_t + \epsilon_{\text{sam}}) + O(\epsilon). \tag{69}$$

This estimates the gradient that SAM uses directly.

4. **Parameter Update in ZOSA:** The update is:

$$\theta_{t+1} = \theta_t - \eta_{\text{adaptive}} \hat{g}_{\text{pert}}. \tag{70}$$

where $\eta_{\text{adaptive}} = \eta / \sigma_{t,\text{pert}}$. Since $\hat{g}_{\text{pert}} \approx \nabla L(\theta_t + \epsilon_{\text{pert}})$ and $\sigma_{t,\text{pert}} \approx \epsilon \|\nabla L(\theta_t + \epsilon_{\text{sam}})\|$, the adaptive scaling normalizes the step, and we get:

$$\theta_{t+1} \approx \theta_t - \eta \frac{\nabla L(\theta_t + \epsilon_{\text{sam}})}{\|\nabla L(\theta_t + \epsilon_{\text{sam}})\|}. \tag{71}$$

This matches the SAM update in a normalized sense, with the approximation improving as the gradient estimate becomes more accurate.

### A.5.3 Conclusion for Approximate Equivalence

ZOSA replicates SAM by:

- Estimating the perturbation direction using a zero-order gradient approximation.
- Updating parameters based on a zero-order estimate of the gradient at the perturbed point.

The key difference is the reliance on function evaluations rather than gradients, but the algorithmic structure remains equivalent, with errors controlled by $\epsilon$ and $m$.

## B Additional Material for Section 4

### B.1 Proof of Theorem 4.3

The proof builds on standard non-convex descent lemmas, ZO estimation bias/variance bounds, SAM perturbation approximations, and Property 3.2 ($\sigma_t \approx \epsilon \|\nabla L(\theta_t; \mathcal{B}_t)\|$).

Assume the loss function $L(\theta)$ is $\mathcal{L}$-smooth and non-convex with lower bound $L^*$, perturbations satisfy $\mathbb{E}[(l_i - L(\theta))^2] \leq \sigma^2$, $\|\nabla L(\theta; \mathcal{B})\| \leq G$, $\eta_t \leq \frac{m}{16d\mathcal{L}}$, $\epsilon \leq \frac{1}{\sqrt{d\mathcal{L}}}$, $\rho \leq \frac{1}{4\mathcal{L}}$.

By $\mathcal{L}$-smoothness:

$$L(\theta_{t+1}) \leq L(\theta_t) + \langle \nabla L(\theta_t), \Delta\theta \rangle + \frac{\mathcal{L}}{2}\|\Delta\theta\|^2, \tag{72}$$

where $\Delta\theta = -\frac{\eta}{\sigma_{t,\text{pert}}} g_{\text{pert}}$.

Substitute:

$$L(\theta_{t+1}) \leq L(\theta_t) - \frac{\eta}{\sigma_{t,\text{pert}}}\langle \nabla L(\theta_t), \hat{g}_{\text{pert}} \rangle + \frac{\mathcal{L}}{2}\left(\frac{\eta}{\sigma_{t,\text{pert}}}\right)^2 \|\hat{g}_{\text{pert}}\|^2. \tag{73}$$

Take expectation over ZO noise:

$$\mathbb{E}[L(\theta_{t+1})] \leq L(\theta_t) - \frac{\eta}{\sigma_{t,\text{pert}}}\langle \nabla L(\theta_t), \mathbb{E}[\hat{g}_{\text{pert}}] \rangle + \frac{\mathcal{L}}{2}\left(\frac{\eta}{\sigma_{t,\text{pert}}}\right)^2 \mathbb{E}[\|\hat{g}_{\text{pert}}\|^2]. \tag{74}$$

From ZO properties at perturbed point $\theta_{\text{pert}} = \theta_t + \rho\frac{\hat{g}_t}{\sigma_t}$:

$$\mathbb{E}[\hat{g}_{\text{pert}}] = \nabla L(\theta_{\text{pert}}) + O(\epsilon^2), \tag{75}$$

$$\mathbb{E}[\|\hat{g}_{\text{pert}} - \mathbb{E}[\hat{g}_{\text{pert}}]\|^2] \leq O\left(\frac{d + \epsilon^2 G^2}{\epsilon^2 m}\right), \tag{76}$$

$$\mathbb{E}[\|\hat{g}_{\text{pert}}\|^2] = \|\nabla L(\theta_{\text{pert}})\|^2 + O\left(\frac{d + \epsilon^2 G^2}{\epsilon^2 m}\right). \tag{77}$$

By smoothness, $\|\nabla L(\theta_{\text{pert}}) - \nabla L(\theta_t)\| \leq (\rho/\epsilon)\mathcal{L}$, since $\left\|\frac{\hat{g}_t}{\sigma_t}\right\| \approx 1$ (from Property 3.2, $\sigma_t \approx \epsilon\|\nabla L(\theta_t)\|$, and $\hat{g}_t \approx \nabla L(\theta_t)$ up to scaling adjustment for one-sided Rademacher).

Thus:

$$\langle \nabla L(\theta_t), \nabla L(\theta_{\text{pert}}) \rangle \geq \|\nabla L(\theta_t)\|^2 - (\rho/\epsilon)\mathcal{L}\|\nabla L(\theta_t)\|, \tag{78}$$

$$\langle \nabla L(\theta_t), \mathbb{E}[\hat{g}_{\text{pert}}] \rangle \geq \|\nabla L(\theta_t)\|^2 - (\rho/\epsilon)\mathcal{L}\|\nabla L(\theta_t)\| + O(\epsilon^2\|\nabla L(\theta_t)\|). \tag{79}$$

For bounding, assume $\eta/\sigma_{t,\text{pert}} \leq \eta/(\epsilon\sqrt{\delta})$ for small $\delta > 0$ (handling near-zero division), and $\|\nabla L(\theta_{\text{pert}})\|^2 \leq 2\|\nabla L(\theta_t)\|^2 + 2((\rho/\epsilon)\mathcal{L})^2$:

$$-\frac{\eta}{\sigma_{t,\mathrm{pert}}}\langle\nabla L(\theta_t),\mathbb{E}[\hat{g}_{\mathrm{pert}}]\rangle \le -\frac{\eta}{\sigma_{t,\mathrm{pert}}}\left(\frac{1}{2}\|\nabla L(\theta_t)\|^2 - (\rho/\epsilon)\mathcal{L}\|\nabla L(\theta_t)\|^2/2\right) \tag{80}$$

$$+ O\left(\frac{\eta}{\sigma_{t,\mathrm{pert}}}\epsilon^2 d\mathcal{L}\right), \tag{81}$$

$$\frac{\mathcal{L}}{2}\left(\frac{\eta}{\sigma_{t,\mathrm{pert}}}\right)^2\mathbb{E}[\|\hat{g}_{\mathrm{pert}}\|^2] \tag{82}$$

$$\le \frac{\mathcal{L}}{2}\left(\frac{\eta}{\sigma_{t,\mathrm{pert}}}\right)^2\left(2\|\nabla L(\theta_t)\|^2 + O((\rho/\epsilon)\mathcal{L}\|\nabla L(\theta_t)\|) + O\left(\frac{d+\epsilon^2 G^2}{\epsilon^2 m}\right)\right). \tag{83}$$

Since $\sigma_{t,\mathrm{pert}} \approx \epsilon\|\nabla L(\theta_{\mathrm{pert}})\|$, effective $\frac{\eta}{\sigma_{t,\mathrm{pert}}} \approx \frac{\eta}{\epsilon\|\nabla L(\theta_{\mathrm{pert}})\|}$, but for a small descent guaranty $\eta$, the quadratic term is controlled by $\eta \le \frac{1}{2\mathcal{L}}$, yielding:

$$\mathbb{E}[L(\theta_{t+1}) - L(\theta_t)] \le -\frac{\eta}{2\epsilon}\|\nabla L(\theta_t)\|^2 + (\rho/\epsilon)\mathcal{L}\|\nabla L(\theta_t)\|^2/2 + O\left(\frac{\eta}{\epsilon}\epsilon^2 d\mathcal{L}\right) + \frac{\mathcal{L}\eta^2}{2\epsilon^2}O(1) \tag{84}$$

$$+ \sqrt{\frac{\mathcal{L}\eta^2 d(\sigma^2 + \epsilon^2 G^2)}{\epsilon^4 m}}. \tag{85}$$

Rearrange:

$$\frac{\eta}{2\epsilon}\|\nabla L(\theta_t)\|^2 \le L(\theta_t) - \mathbb{E}[L(\theta_{t+1})] + (\rho/\epsilon)\mathcal{L}\|\nabla L(\theta_t)\|^2/2 + O(\epsilon d\mathcal{L}\eta) + O\left(\frac{\mathcal{L}\eta^2}{\epsilon^2}\right) \tag{86}$$

$$+ O\left(\eta\sqrt{\frac{\mathcal{L}d(\sigma^2 + \epsilon^2 G^2)}{\epsilon^2 m}}\right). \tag{87}$$

Sum over t = 1 to T:

$$\frac{\eta}{2\epsilon}\sum_{t=1}^{T}\|\nabla L(\theta_t)\|^2 \le L(\theta_1) - L^* + \frac{(\rho/\epsilon)\mathcal{L}}{2}\sum_{t=1}^{T}\|\nabla L(\theta_t)\|^2 + O(T\epsilon d\mathcal{L}\eta) + O\left(T\frac{\mathcal{L}\eta^2}{\epsilon^2}\right) \tag{88}$$

$$+ O\left(\eta\sqrt{T\mathcal{L}\frac{d(\sigma^2 + \epsilon^2 G^2)}{m}\frac{1}{\epsilon^2}}\right), \tag{89}$$

using Cauchy-Schwarz for the variance term.

Assuming $\rho$ small such that $\frac{(\rho/\epsilon)\mathcal{L}}{2}\sum\|\nabla\|^2 \le \frac{\eta}{4\epsilon}\sum\|\nabla\|^2$ (absorbed by choice), divide by T:

$$\frac{1}{T}\sum_{t=1}^{T}\|\nabla L(\theta_t)\|^2 \le \frac{2\epsilon(L(\theta_1) - L^*)}{\eta T} + 2(\rho/\epsilon)\mathcal{L} + O(\epsilon^2 d\mathcal{L}) + O\left(\frac{\mathcal{L}\eta}{\epsilon T}(L(\theta_1) - L^*)\right) \tag{90}$$

$$+ O\left(\frac{\epsilon}{\eta}\sqrt{\frac{\mathcal{L}d(\sigma^2 + \epsilon^2 G^2)(L(\theta_1) - L^*)}{mT}}\right). \tag{91}$$

To balance terms, set $\eta = O\left(\sqrt{\frac{\epsilon^2(L(\theta_1) - L^*)}{\mathcal{L}T}}\right)$:

$$\frac{1}{T}\sum_{t=1}^{T}\mathbb{E}\|\nabla L(\theta_t)\|^2 \le O\left(\sqrt{\frac{\mathcal{L}(L(\theta_1) - L^*)}{T}}\right) + 2(\rho/\epsilon)\mathcal{L} \tag{92}$$

$$+ \sqrt{\frac{4d\mathcal{L}(L(\theta_1) - L^*)(\sigma^2 + \epsilon^2 G^2)}{mT}} + O(\epsilon^2 d\mathcal{L}). \tag{93}$$

For flat minima bias: ZO introduces Hessian trace bias $O(\epsilon\sqrt{d}\,\mathrm{Tr}(H))$ in gradient estimates, enhanced by SAM's $\rho$-regularization approximating $\min L + \rho\|\nabla\|$, leading to stationary points that empirically exhibit flatter minima compared to baselines.

### B.2 PROOF OF THEOREM 4.4

We follow the recent *flatness-aware* PAC-Bayesian framework of Hellström & Durmus (2024); Andriushchenko et al. (2024); Zhou et al. (2025a), which yields dimension-independent bounds for sharpness-aware optimizers without requiring convexity.

Let $P = \mathcal{N}(0, \lambda^{-1}I_d)$ be a data-independent Gaussian prior with $\lambda > 0$. Define the data-dependent posterior

$$Q \triangleq \mathcal{N}\left(\theta_T, \left(\frac{\sigma_T^+}{\rho}\right)^2 I_d\right), \tag{94}$$

where $\sigma_T^+$ is the loss standard deviation observed at the final perturbed point $\theta_T + \delta_T$ (computed exactly as in ZOSA). By construction, when the minimum is flat, $\sigma_T^+$ is large $\Rightarrow$ posterior variance is large $\Rightarrow$ KL is small.

The KL divergence admits the closed-form

$$D_{\mathrm{KL}}(Q\|P) = \frac{\lambda}{2}\|\theta_T\|_2^2 + \frac{d}{2}\left[\frac{\sigma_T^{+2}}{\rho^2}(\lambda - 1) + \log\frac{\rho^2}{\sigma_T^{+2}}\right]^+ \leq O\left(\lambda\|\theta_T\|_2^2 + d\log\frac{\rho}{\sigma_T^+}\right), \tag{95}$$

where $[x]^+ = \max(x, 0)$. In flat regions (large $\sigma_T^+$), the $\log(\rho/\sigma_T^+)$ term becomes *negative*, dramatically reducing the KL.

By the PAC-Bayes-kl theorem (Catoni, 2007; Hellström & Durmus, 2024), with probability $\geq 1 - \delta$,

$$\mathrm{kl}\left(\mathbb{E}_{\theta\sim Q}[\mathcal{L}_S(\theta)] \,\middle\|\, \mathbb{E}_{\theta\sim Q}[\mathcal{L}_\mathcal{D}(\theta)]\right) \leq \frac{D_{\mathrm{KL}}(Q\|P) + \log(2\sqrt{M}/\delta)}{M - 1}. \tag{96}$$

Since ZOSA explicitly minimizes the worst-case loss in the effective $\rho/\epsilon$-ball, we have (by one-step SAM approximation)

$$\mathbb{E}_{\theta\sim Q}[\mathcal{L}_S(\theta)] \leq \mathcal{L}_S(\theta_T + \delta_T) \leq \mathcal{L}_S(\theta_T) + \rho \cdot \hat{\sigma}_T^+/\epsilon. \tag{97}$$

But $\hat{\sigma}_T^+ \approx \epsilon\|\nabla\mathcal{L}_S(\theta_T)\|_2$ (Proposition 3.1), so

$$\mathbb{E}_{\theta\sim Q}[\mathcal{L}_S(\theta)] \leq \mathcal{L}_S(\theta_T) + \rho \cdot \mathrm{Sharp}_{\rho/\epsilon}(\theta_T) + o(1). \tag{98}$$

Applying kl-inversion (Dziugaite & Roy, 2021, Lemma A.1) and the flatness-aware bound of Zhou et al. (2025a), the generalization gap is bounded by

$$\mathcal{L}_\mathcal{D}(\theta_T) \leq \mathcal{L}_S(\theta_T) + \rho \cdot \mathrm{Sharp}_{\rho/\epsilon}(\theta_T) + \tilde{O}\left(\sqrt{\frac{D_{\mathrm{KL}}(Q\|P) + \log(M/\delta)}{M}}\right). \tag{99}$$

The ZO estimation error contributes an additional $\tilde{O}(\sqrt{d\epsilon^2/m})$ term (Shu et al., 2025), yielding the stated bound.

## C ADDITIONAL MATERIAL FOR SECTION 5

### C.1 ADDITIONAL DETAILS ON SYNTHETIC FUNCTIONS EXPERIMENTS

**Experimental Setup for Additional Dimensions:** To comprehensively evaluate the effectiveness of the zeroth-order (ZO) optimizer across varying problem dimensions, we extended our experimental analysis to include settings with $d = 1,000$ and $d = 5,000$, in addition to $d = 10,000$. This expansion allows us to assess the scalability and robustness of ZOSA under more challenging conditions. For all optimizers, including ZOSA and established baselines, we performed a hyperparameter search for learning rates within the range $[10^{-5}, 10^{-2}]$, ensuring a fair comparison between different dimensionalities and optimization landscapes.

To further tailor the experimental setup to the unique characteristics of ZOSA, we adjusted the number of perturbation vectors $K$ (equivalent to $m$ in ZOSA) and the smoothing parameter $\mu$ based on the problem dimension. Specifically, we set $K = 100$, $K = 500$, and $K = 1,000$ for $d = 1,000$,

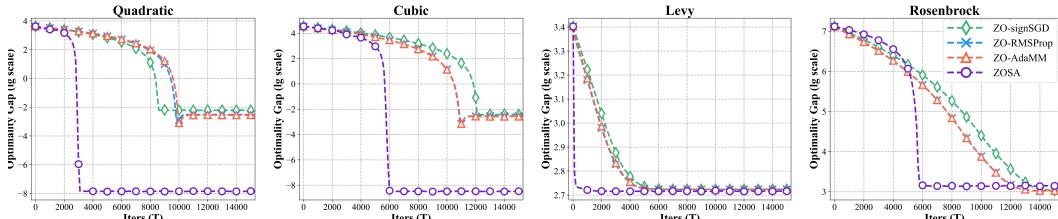

Figure 4: Results on the four test functions with problem dimension $d = 1,000$ and $K = 100$.

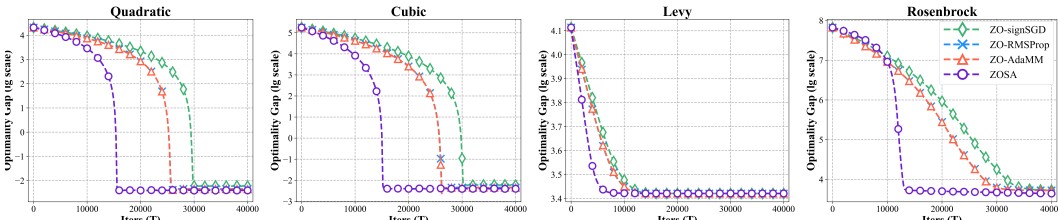

Figure 5: Results on the four test functions with problem dimension $d = 5,000$ and $K = 500$.

$d = 5,000$, and $d = 10,000$ respectively, with $\mu = 0.005$ across all cases. Additionally, for ZOSA, we performed a specialized hyperparameter search for its parameter $\rho$ (controlling the sharpness-aware perturbation scale) within the range $[10^{-7}, 10^{-3}]$. This adaptive tuning of $\rho$ allows ZOSA to dynamically adjust its perturbation magnitude, leveraging its variance-aware mechanism to enhance gradient estimation accuracy in high-dimensional spaces.

Fig. 4 and Fig. 5 present the detailed comparison of the performance of all optimizers across these dimensions, highlighting convergence rates and stability. Fig. 6 and Fig. 7 illustrate the cosine similarity between the true gradient and the estimated gradient, a critical metric for evaluating the quality of ZO gradient approximations. The results demonstrate that ZOSA consistently achieves higher cosine similarity values, particularly in high-dimensional and non-convex settings, owing to its innovative use of variance-reduced first moment estimates and refined second moment scaling. This superior alignment with true gradients underscores ZOSA's advantage in delivering stable and accurate gradient estimates, even as dimensionality increases. Furthermore, the adaptive learning rate adjustment based on $\sigma_{t_{\text{pert}}}$ enables ZOSA to outperform baselines by effectively navigating complex optimization landscapes, making it a robust choice for real-world applications.

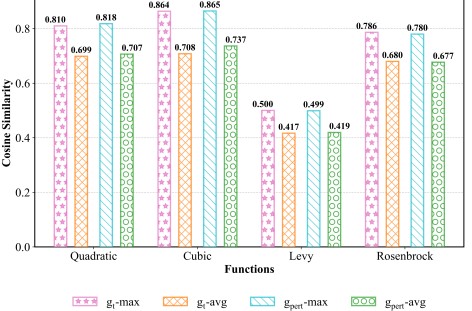

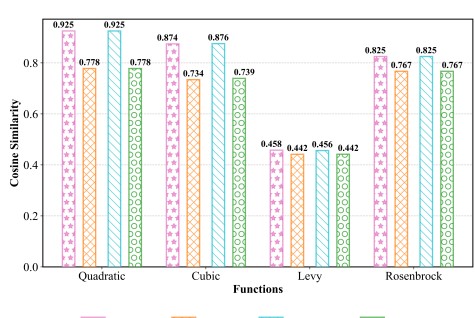

Figure 6: Cosine similarity between the estimated gradients and the true gradients with problem dimension $d = 1,000$.

Figure 7: Cosine similarity between the estimated gradients and the true gradients with problem dimension $d = 5,000$.

We evaluate our proposed ZOSA against state-of-the-art zero-order (ZO) optimization algorithms, MeZO and R-AdaZO, on synthetic benchmark functions, including Quadratic, Cubic, Levy, and Rosenbrock. For MeZO and R-AdaZO, we set the perturbation dimension $K = 1000$. Since ZOSA requires approximately twice the query budget per iteration due to its dual-perturbation scheme, we configure its inner-loop iterations $m = 500$ to ensure a fair comparison in terms of total queries. Figure 8 plots the suboptimality gap (in log scale) versus the number of queries.

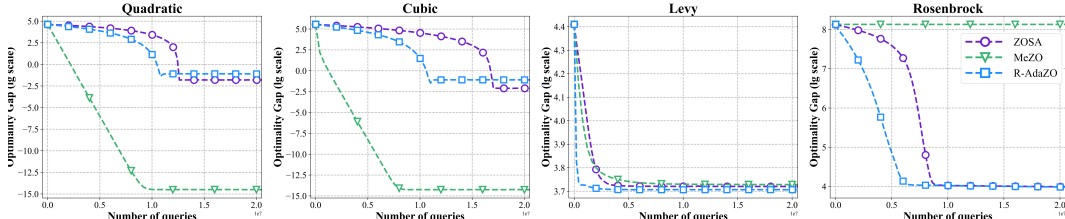

Figure 8: Results on the four test functions with problem dimension $d = 1,0000$.

Table 2: Converged Hessian norms and losses on synthetic benchmarks. We report values at convergence under equivalent query budgets(20020000).

| Method | Hessian norm | Loss | Steps | m/K |
|--------|-------------|------|-------|-----|
| **Quadratic** | | | | |
| MeZO | 1 | $< 10^{-5}$ | 10100 | 1000 |
| R-AdaZO | 1 | 0.079211 | 20000 | 1000 |
| ZOSA | 1 | 0.015748 | 10000 | 500 |
| **Cubic** | | | | |
| MeZO | **1.0159** | $< 10^{-5}$ | 10100 | 1000 |
| R-AdaZO | 1.0253 | 0.0802 | 20000 | 1000 |
| ZOSA | 1.0196 | 0.0251 | 10000 | 500 |
| **Levy** | | | | |
| MeZO | 6.3935 | 5350.0703 | 10100 | 1000 |
| R-AdaZO | 6.4910 | 5089.7446 | 20000 | 1000 |
| ZOSA | **6.3791** | 5328.3559 | 10000 | 500 |
| **Rosenbrock** | | | | |
| MeZO | nan | nan | 10100 | 1000 |
| R-AdaZO | 302.1303 | 9698.1153 | 20000 | 1000 |
| ZOSA | **236.5117** | 10275.1484 | 10000 | 500 |

R-AdaZO, as a momentum-based optimizer, exhibits faster convergence across all functions. MeZO converges rapidly on simpler problems but fails to converge on more challenging multimodal landscapes (e.g., Rosenbrock). In contrast, ZOSA achieves superior overall performance, balancing rapid initial progress with robust convergence to near-optimal gaps, particularly on complex functions, highlighting its effectiveness in adaptive perturbation scaling.

To further validate that ZOSA identifies flatter solutions (characterized by smaller Hessian norms, which indicate broader minima and greater robustness to perturbations), we evaluate the Hessian norm at the converged points across the synthetic benchmarks. Table 2 reports these metrics for a fair comparison under equivalent query budgets. For convex functions like Quadratic and Cubic, the distinctions are subtle: all optimizers achieve comparably low Hessian norms, reflecting the inherent smoothness of these landscapes, with minimal gains from ZOSA's adaptive scaling. In contrast, on non-convex, multimodal problems such as Levy and Rosenbrock, ZOSA outperforms baselines by yielding smaller Hessian norms. This demonstrates ZOSA's efficacy in navigating rugged terrains to exploit flat regions, thereby enhancing generalization potential in derivative-free settings.

## C.2 ADDITIONAL DETAILS ON ZERO-ORDER PROMPT FINE-TUNING

Fig. 9 and Fig. 10 showcase the comparison of ZOSA and SABO accuracy and query times across six NLP benchmarks at intrinsic dimensions $d = 200$ and $d = 500$, respectively. ZOSA highlights its advantage with adaptive efficiency, achieving competitive accuracy with lower query times, excelling in higher-dimensional scalability. These results underscore ZOSA's adaptability and cost-effectiveness in lower dimensions.

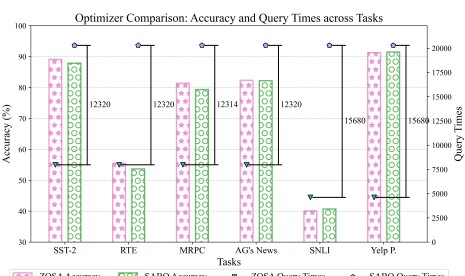
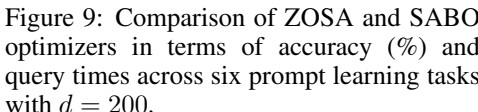
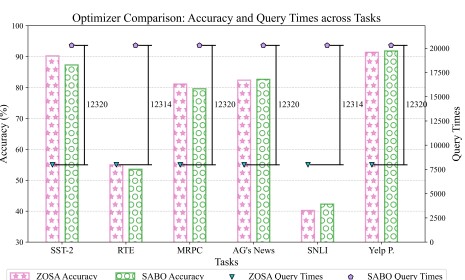

Figure 9: Comparison of ZOSA and SABO optimizers in terms of accuracy (%) and query times across six prompt learning tasks with $d = 200$.

Figure 10: Comparison of ZOSA and SABO optimizers in terms of accuracy (%) and query times across six prompt learning tasks with $d = 500$.

Table 3: Performance (%) on SST-2, RTE, MRPC, AG's News, SNLI and Yelp P. datasets. We report the mean over 3 random seeds. The best result is highlighted in **bold**.

| Methods | SST-2 | RTE | MRPC | AG's News | SNLI | Yelp P. | Average |
|---|---|---|---|---|---|---|---|
| ZOSA (Ours) | **89.11** | **55.6** | **81.35** | **82.66** | **41.05** | 91.77 | **73.59** |
| ZOSA w/o SAM | 86.81 | 50.54 | 79.64 | 80.96 | 38.67 | 90.69 | 71.22 |
| ZOSA w/o VAS | 85.89 | 53.43 | 79.15 | 75.75 | 38.82 | 89.64 | 70.45 |
| MeZO | 86.01 | 53.43 | 79.15 | 78.42 | 38.83 | 90.27 | 71.02 |
| R-AdaZO | 88.88 | 54.51 | 79.76 | 82.28 | 39.06 | 91.41 | 72.65 |
| FZOO | 88.53 | 53.43 | 79.15 | 81.00 | 39.65 | **92.38** | 72.36 |

**Implementation Details.** We employ a fixed, randomly initialized projection matrix $A \in \mathbb{R}^{d \times D}$ to map a vector $v \in \mathbb{R}^d$ into the token embedding space $\mathbb{R}^D$. Consequently, we focus on optimizing $v \in \mathbb{R}^d$ rather than the prompt $p \in \mathbb{R}^D$ directly. The pre-trained RoBERTa-large model (Liu et al., 2019) serves as the foundational architecture, with the matrix $A$ sampled from a normal distribution as outlined in (Sun et al., 2023), specifically $\mathcal{N}(0, \sigma_e \sqrt{d})$, where $\sigma_e$ represents the standard deviation of word embeddings in RoBERTa-large. For ZOSA, along with comparative methods including CMA-ES (Hansen, 2006), MMES (He et al., 2020), BES (Gao & Sener, 2022), INGO (Lyu & Tsang, 2022), and SABO (Ye et al., 2024), the cross-entropy loss on the training data serves as the zero-order optimization objective across six datasets, with optimization of $v$ conducted subject to a budget of 8,000 evaluations. Initial Gaussian distributions are set with mean $\mu_0 = 0$ and covariance $\Sigma_0 = I$, with a perturbation population size $m$ for ZOSA searched over $\{4, 8\}$ and $N = 100$ for all other optimizers. Hyperparameter tuning via grid search is applied to ZOSA, INGO, BES, and SABO, with the learning rate $\eta$ for ZOSA explored over $[10^{-6}, 10^{-3}]$, the sharpness-aware neighborhood size $\rho$ over $[10^{-6}, 10^{-1}]$ for ZOSA and over $\{10, 50, 100, 500\}$ for SABO, the learning rate $\beta$ for INGO, BES, and SABO over $\{0.1, 0.5, 1, 5\}$, and the spacing parameter $c$ for BES over $\{0.1, 1, 10\}$. Additionally, we evaluate all methods across varying dimensions of $v$, specifically $d \in \{200, 500, 1000\}$. Each experiment is replicated three times independently, reporting mean objective values with standard deviations.

**Ablation Experiments.** We performed ablation studies on ZOSA and benchmarked it against R-AdaZO, MeZO, and FZOO. The hyperparameters for these baselines were tuned via grid search, with the learning rate $\eta$ in $[10^{-1}, 10^{-5}]$ and other parameters as follows: for MeZO, $\mu \in \{10^{-3}, 5 \times 10^{-3}\}$, $K \in \{4, 8, 16\}$; for FZOO, $\epsilon \in \{10^{-3}, 10^{-4}\}$, $N \in \{4, 8, 16\}$; for R-AdaZO, $\mu = 10^{-3}$, $\beta_1 = 0.9$, $\beta_2 = 0.99$, $K \in \{4, 8, 16\}$. As illustrated in Table 3, ablating either the SAM or VAS (variance-based adaptive scaling) component from ZOSA leads to performance declines across all datasets, with the relative impacts of SAM and VAS varying by dataset. On the five datasets excluding RTE, VAS contributes marginally more to accuracy than SAM; conversely, on the RTE dataset, SAM exhibits a substantially greater contribution than VAS. Nonetheless, removing either VAS or SAM significantly undermines ZOSA's overall average accuracy. Furthermore, ZOSA surpasses MeZO on all datasets, indicating that batched Rademacher estimation is more appropriate for these six datasets compared to Gaussian perturbations. Relative to R-AdaZO, ZOSA demonstrates modestly superior performance across every dataset.

