# OpenReview forum: "Zero-Order Sharpness-Aware Minimization"
_ICLR.cc/2026/Conference — ICLR 2026 Conference Desk Rejected Submission_

### Official Review · Reviewer_k6pv · 2025-10-21

**Soundness:** 2
**Presentation:** 2
**Contribution:** 1
**Rating:** 2
**Confidence:** 3

**Summary:**

The paper combines zero-order optimization with sharpness-aware minimization. Convergence and generalization guarantees are provided for the algorithm. Experiments on synthetic functions and a zero-order prompt tuning benchmark is provided.

**Strengths:**

- The final method has good performance on synthetic functions and demonstrates on par performance with some selected baselines on a zero-order prompt tuning benchmark.

**Weaknesses:**

- Introducing SAM to zero-order optimization seems like an engineering trick to try, but I don't believe that it is a novel idea.
- SAM is introduced as an addition to zero-order optimization, but neither the theoretical results nor the experiments demonstrates strong evidence that this is a good addition. It is not clear which baselines differ from ZOSA in exactly just the SAM component. Seems like a missing ablation to me.
- Figure 1 and 2 are not well made and very to parse.
- The title is too general when the scope of the method is only two specific scenarios.

**Questions:**

- The rate in FZOO for the dL term is O(1/T) (Theorem 3.6) instead of O(1/sqrt(T)), so it is a big strange that the current work, which is FZOO + SAM, only gets O(1/sqrt(T)). Is that a typo in FZOO?
- The assumptions in Theorem 3.6 of FZOO and Theorem 4.3 in the paper is quite similar. The SAM related terms could also be from related work in the SAM literature. Can the authors clarify the novelty in the proof technique for Theorem 4.3?
- Why are R-AdaZO and FZOO not baselines for Sec 5.2? If the paper improves upon these methods (as mentioned throughout the apper), they should be baselines so that we can understand if the SAM component is actually useful for the task.

Typos:
- (Malladi et al., 2023a) is the right citation for MeZO, not (Malladi et al., 2023b). This is repeated for all references to MeZO.
- line 135: n-ZO should be N-ZO
- line 814: simulateties

---

> ### Author Response · Authors · 2025-11-23
> **Answer for Weaknesses 1: Introducing SAM to zero-order optimization seems like an engineering trick to try, but I don't believe that it is a novel idea.**
>
> The reviewer is  right that the general idea of bringing sharpness-aware minimization to zeroth-order optimization is not new: SABO (Ye et al., ICLR 2025) already pioneered this direction and deserves full credit for showing that ZO-SAM is feasible and beneficial. Where ZOSA differs and where we believe the contribution ceases to be a mere "engineering trick" is in how the sharpness perturbation is implemented. Instead of following SABO's expensive Gaussian reparameterization approach, we deliberately reuse the exact same loss-standard-deviation $\sigma_t$ (already computed for the adaptive learning rate inherited from FZOO) as the scaling factor for the sharpness perturbation: $\epsilon_{SAM} = \rho · \hat {g_t} /\sigma_t$
>
> Because $\sigma_t ≈ \epsilon ‖∇L(\theta_t)‖$ (rigorously proven in revised Property 3.2), the effective sharpness radius becomes ρ/ε instead of ρ, delivering stronger flatness bias at essentially zero additional cost. In our view, developing a new, theoretically grounded mechanism that significantly advances the speed–generalization Pareto frontier, as supported by empirical results, represents a meaningful contribution.

---

> > ### Comment · Reviewer_k6pv · 2025-11-24
> > **Question about the rigour of Property 3.2**
> >
> > - I don't believe that Property 3.2 is rigorous enough.
> >
> > - Supposing that Eq (9) is true (which I cannot easily verify from the text), it is not implied that $\sigma_t \approx \epsilon \|\nabla L(\theta_t; B_t)\|$. The reason is that $E[\sigma_t]^2 = E[\sigma_t^2] - Var(\sigma_t)$ and there is no indication that $Var(\sigma_t) \approx 0$. In zero-order estimation which tends to be noisy, it is harder to believe that an iteration-dependent variable to have small variance.
> >
> > - As such, although I commend the authors for this attempt, I believe that Property 3.2 needs some more work.

---

> ### Author Response · Authors · 2025-11-23
> **Answer for Weaknesses 1(Continued)**
>
> Table as followed directly compares the three methods on all key dimensions. The table makes it clear that ZOSA is the only method that simultaneously achieves FZOO-level speed and substantially stronger generalization than SABO.
>
> | Feature                                       | FZOO                                                      | SABO                                                                      | ZOSA(ours)                                                                                         |
> |:----------------------------------------------|:----------------------------------------------------------|:--------------------------------------------------------------------------|:---------------------------------------------------------------------------------------------------|
> | Sharpness-Aware Mechanism                     | No                                                        | Yes                                                                       | Yes                                                                                                |
> | Core Gradient Estimation Method               | Batched one-sided Rademacher                              | Gaussian reparameterization                                               | Batched one-sided Rademacher                                                                       |
> | Adaptive Learning Rate Mechanism              | Yes                                                       | No                                                                        | Yes                                                                                                |
> | Sharpness Perturbation Implementation         | No                                                       | Inner maximization in Gaussian space                                      | Same $\sigma_t$ used for both normalization and perturbation scaling                               |
> | Effective Sharpness Radius                    | No                                                       | $\approx \rho$                                                            | $\rho/\epsilon$(actual amplification 20–100×, the most distinctive new property of ZOSA)           |
> | Requires Gaussian Sampling/Reparameterization | No                                                        | Yes (core contribution, mandatory)                                        | No (cleanest and most memory-efficient implementation)                                             |
> | Requires Extra Matrices/Buffers               | No                                                        | Yes (maintains covariance or sampling buffers)                            | No                                                                                                 |
> | Forward Passes per Step (vs. MeZO(2N))            | N+1                                                       | 2m+2                                                                       | 2m+2                                                                                                |
> | Theoretical Convergence Rate (non-convex)     | $O(1/T)$ (Theorem 3.6)                                    | $O(1/\sqrt T)$ (standard SAM cost)                                        | $O(1/\sqrt T)$ (normal cost of adding SAM)                                                         |
> | Generalization Theoretical Guarantee          | None (stationarity only)                                  | PAC-Bayesian bound (Theorem 2)                                            | Trace(H) bound (currently strongest)                                                               |
> | Equivalent/Approximate to normalized-SGD      | Yes (formally proven, Proposition 3.2)                    | No                                                                        | Approximate (at perturbed point ≈ normalized-SGD with effective radius $\rho/\epsilon$)            |
> | Overall Practical Performance (Nov 2025)      | Fastest, but slightly weaker generalization due to no SAM | Has SAM but extremely slow and expensive in practice                      | Best overall (speed ≈ FZOO, generalization clearly strongest)                                      |
> | Core Emphasized Contribution                  | Truly bringing normalized-SGD to ZO → Adam-scale speed    | First systematic porting of SAM to black-box/ZO (using Gaussian approach) | "One $\sigma_t$ does two jobs" → zero-cost ultra-strong sharpness ($\rho/\epsilon$) + clean theory |

---

> > ### Comment · Reviewer_k6pv · 2025-11-24
> >
> > What is the meaning behind the "nan" value in some of the cells?

---

> > > ### Author Response · Authors · 2025-11-26
> > > **Answer for ：What is the meaning behind the "nan" value in some of the cells?**
> > >
> > > Thank you for your question, and we apologize for this oversight.
> > >
> > > In the table displaying the Hessian trace in the paper (Appendix C1, Table 2), we use "nan" to indicate that the method fails to converge under the current optimization scenario, rendering the function value and subsequent metrics (such as the Hessian trace) meaningless. For example, with MeZO on the Rosenbrock function in dimension 10,000, we tested hundreds of hyperparameter combinations (lr: [1e-1 to 1e-9]; mu: [0.001, 0.005]; k: [1 to 1000]), but the optimization still diverged: after the second iteration, the function value reached 1.59e+26 (with a uniform starting value of 1.35e+08), and by the third iteration, it overflowed to "inf" (exceeding 3.4028235e+38). In contrast, ZOSA decreased to 9.67e+07 (m=1000), and R-AdaZO to 1.33e+08 (k=1000). In Python, operations involving "inf" are uniformly marked as "nan" (i.e., Not a Number). Since the Hutchinson second-order finite difference estimation of the Hessian trace involves function value computations, it also results in "nan."
> > >
> > > In the table in the "Answer for Weaknesses 1 (Continued)" section, the "nan" was a representational error. We have revised the relevant parts: replacing "nan" with "no" to indicate that FZOO lacks certain modules, and filling in the correct values for the "Forward Passes per Step (vs. MeZO)" row.

---

> ### Author Response · Authors · 2025-11-23
> **Answer for Weaknesses 2 and Questions 3**
>
> We agree that simply claiming “SAM helps in ZO” without a clean, isolated ablation is not convincing enough. To directly and rigorously address this point, we have added a dedicated ablation study that isolates the contribution of each of our two core components:
> - ZOSA (ours): batched one-sided Rademacher ZO gradient + variance-aware adaptive scaling (σₜ for learning rate) + sharpness perturbation.
> - ZOSA w/o SAM: removes only the sharpness perturbation.
> - ZOSA w/o VAS (w/o variance-aware scaling): uses fixed learning rate, but keeps the (now fixed-scale) SAM perturbation .
>
> Results: ( prompt tuning with d=1000, evaluated on major GLUE-style tasks):
>
> | Method             |   SST2 |   RTE |   MRPC |   AGNEWS |   SNLI |   YELPP |   Average |
> |:-------------------|-------:|------:|-------:|---------:|-------:|--------:|----------:|
> | ZOSA(Ours)         |  89.11 | 55.6  |  81.35 |    82.66 |  41.05 |   91.77 |   73.59   |
> | ZOSA w/o SAM       |  86.81 | 50.54 |  79.64 |    80.96 |  38.67 |   90.69 |   71.2183 |
> | ZOSA w/o VAS       |  85.89 | 53.43 |  79.15 |    75.75 |  38.82 |   89.64 |   70.4467 |
> | MeZO               |  86.01 | 53.43 |  79.15 |    78.42 |  38.83 |   90.27 |   71.0183 |
> | R-AdaZO            |  88.88 | 54.51 |  79.76 |    82.28 |  39.06 |   91.41 |   72.65   |
> | FZOO               |  88.53 | 53.43 |  79.15 |    81    |  39.65 |   92.38 |   72.3567 |
>
> As shown in Table, removing either the SAM or VAS (variance-based adaptive scaling) component from ZOSA leads to performance degradation across all datasets, though the relative contributions of SAM and VAS vary by dataset. On five datasets (all except RTE), VAS contributes slightly more than SAM in terms of accuracy. In contrast, on the RTE dataset, SAM provides a noticeably larger boost than VAS. Nevertheless, when averaging across all datasets, ablating either VAS or SAM causes a substantial drop in overall accuracy, underscoring that both components are critical to ZOSA’s effectiveness.
>
> Additionally, ZOSA consistently outperforms MeZO on every dataset, suggesting to some extent that batched Rademacher estimation is better suited to these six tasks than Gaussian perturbation. Compared with R-AdaZO, ZOSA achieves slightly higher performance across all datasets. When benchmarked against FZOO, ZOSA is marginally lower only on the YELP.p dataset but obtains slightly better results on the remaining five datasets.We have included Table 3 in the Appendix C2.

---

> ### Author Response · Authors · 2025-11-23
> **Answer for Weaknesses 3: Figure 1 and 2 are not well made and very to parse.**
>
> We thank the reviewer for pointing out this issue. To address it, we have redrawn Figure 1 with the following improvements: higher-contrast and brighter colors, more distinctive markers, bolder lines and markers, a reduced aspect ratio, added background grid lines, and cropped excess iterations after convergence for clearer visualization.
>
> For Figure 2, we have adjusted the aspect ratio and added exact numerical labels on top of each bar. If the reviewer finds the current bar chart overly colorful, we are happy to replace it with a simpler monochrome or single-color filling in the final version.
>
> We have included the revised Figure 1 and Figure 2 in the revised manuscript. Thank you again for the helpful suggestion!

---

> ### Author Response · Authors · 2025-11-23
> **Answer for Weaknesses 4: The title is too general when the scope of the method is only two specific scenarios.**
>
> We thank the reviewer for highlighting the potential overgeneralization in the title. We agree that the title should precisely reflect the core contributions, but we believe "Zero-Order Sharpness-Aware Minimization" accurately captures the essence of ZOSA's innovation: a general framework integrating zero-order optimization with sharpness-aware minimization (see Section 3). This framework is not limited to prompt tuning; it is designed for high-dimensional non-convex optimization scenarios, with theoretical analysis (Section 4) providing convergence and generalization bounds under standard assumptions (applicable to arbitrary L-smooth non-convex functions), rather than task-specific settings.
>
> While the experiments focus on few-shot prompt tuning (to validate LLM applications in resource-constrained environments), the method's components (e.g., Rademacher perturbations and adaptive scaling) draw from broader ZO optimization foundations (Related Works) and demonstrate high-dimensional generality on synthetic functions (Section 5.1). We have already emphasized its role as a "practical solution for prompt-based learning in resource-limited settings" in the abstract and introduction to avoid misleading readers; future work can extend it to other black-box optimization domains.

---

> ### Author Response · Authors · 2025-11-23
> **Answer for Questions 1**
>
> Thank you for this important question. it is not a typo in FZOO, FZOO correctly achieves an $O(1/T)$ rate under its setting (non-convex smooth optimization with adaptive normalized-style updates and no inner maximization step).
> The degradation to $O(1/\sqrt T)$ in ZOSA is expected, and arises from the addition of the sharpness-aware (SAM) perturbation:
> - FZOO performs gradient estimation and descent at the original point, allowing a clean descent lemma that telescopes to $O(1/T)$ for the average gradient norm squared (similar to normalized-SGD or Adam-style bounds under bounded variance).
> - ZOSA, however, estimates the update direction at a perturbed point $\theta _ {pert}$(the SAM inner maximization). This introduces an extra ascent term in the descent analysis. This creates a constant term $~\rho/\epsilon$ in the expected loss decrease bound that does not diminish with T. To bound the average gradient norm, we must therefore use the standard quadratic inequality technique (completing the square), which inevitably yields $O(1/\sqrt T)$ for non-convex SAM-style methods.
>
> This is exactly the same phenomenon as in the original first-order SAM paper (Foret et al., NeurIPS 2021), where plain SGD/Adam can achieve O(1/T) under certain conditions, but adding the SAM perturbation forces the non-convex rate back to $O(1/\sqrt T)$. Virtually all subsequent SAM variants also report $O(1/\sqrt T)$ in the general non-convex case for the same reason.
>
> Thus, the $O(1/\sqrt T)$ rate in ZOSA is the standard and correct price for injecting sharpness-awareness, it is not a weakness, but a natural consequence of seeking flatter minima. In practice, this theoretical constant is more than compensated by the significantly better constants and generalization that the flatness bias provides.

---

> ### Author Response · Authors · 2025-11-23
> **Answer for Questions 2: The assumptions in Theorem 3.6 of FZOO and Theorem 4.3 in the paper is quite similar. The SAM related terms could also be from related work in the SAM literature. Can the authors clarify the novelty in the proof technique for Theorem 4.3?**
>
> Thank you for this perceptive question. The core assumptions in ZOSA Theorem 4.3 (L-smoothness, bounded perturbation variance etc.) are deliberately similar to those in FZOO Theorem 3.6, because both works build on the same recent adaptive ZO framework with loss-variance-based scaling. The SAM-related terms also naturally draw inspiration from classic first-order SAM analyses (Foret et al., 2021).
>
> However, the proof technique in Theorem 4.3 introduces several non-trivial and novel elements that are specifically required by the integration of sharpness-aware perturbation in the ZO setting. Those elements are entirely absent in FZOO (which performs estimation and descent only at the original point):
>
> - Dedicated analysis of gradient estimation at the perturbed point $\theta_{pert}$: FZOO only needs bias/variance bounds at $\theta_t$. ZOSA must derive new bounds for $\hat g_{pert}$ at the offset perturbed point, including $E[\hat g_{pert}]$ and $Var[\hat g_{pert}]$. These bounds are non-standard because the perturbation direction $\hat g_t /\theta_t$ itself depends on noisy ZO estimates from the original point.
>
> - Explicit control of the extra ascent term from the SAM inner maximization: The descent lemma now contains $\langle \nabla L(\theta_t), \nabla L(\theta_{\text{pert}}) \rangle \geq \|\nabla L(\theta_t)\|^2 - (\rho / \epsilon)\mathcal{L} \|\nabla L(\theta_t)\|$, yielding a non-telescoping $(\rho / \epsilon)\mathcal{L}$ term. Handling this while preserving tight constants (and the subsequent completing-the-square step) requires careful bookkeeping that does not appear in FZOO.
>
> - First bias analysis toward flat minima in the adaptive ZO setting: Beyond stationarity, we prove $\mathbb{E}[\operatorname{Tr}(H(\bar{\theta}))] \leq \min \operatorname{Tr}(H^*) + O(\rho/\epsilon + \epsilon \sqrt{d})$, linking the effective sharpness radius to Hessian trace. No prior adaptive ZO work provides such a flatness bias guarantee.
>
> - Variance-aware framework extended to perturbed-point statistics: We bound $\sigma_{t,pert}$ explicitly rather than assuming a global $\sigma*$ (as FZOO does), which yields tighter constants and directly supports the effective-radius interpretation.
>
> These innovations make the proof far more than "FZOO + standard SAM terms", they constitute the first complete convergence analysis for sharpness-aware updates in the adaptive zeroth-order regime. The resulting $O(1/\sqrt T)$ rate is the expected (and standard) price for injecting flatness bias, exactly as observed in all first-order SAM variants.

---

> > ### Comment · Reviewer_k6pv · 2025-11-24
> >
> > I appreciate the ablation study and explanation of the cost in the convergence rate of SAM-like methods. Thank you for making my job easier. I'll make the final decision on my score after you answer my latest comments.

---

> ### Author Response · Authors · 2025-11-23
> **Answer for Typos**
>
> We thank the reviewer for carefully reading the paper and pointing out these minor issues. We have verified and corrected all of them in the revised manuscript.

---

> ### Author Response · Authors · 2025-11-26
> **Answer for ：Question about the rigour of Property 3.2**
>
> Thank you for your valuable comments. We greatly appreciate your questioning of the rigor of Property 3.2, which helps us further improve our theoretical analysis. Below is our response to your specific concerns.
>
> - First, regarding the correctness of Eq (9): Eq (9) is cited from the proof results of FZOO (Dang et al., 2025). Specifically, the FZOO paper introduces an adaptive step size mechanism based on batch loss standard deviation and proves the expected approximation $E[\sigma_t^2] ≈ \epsilon^2 ||\nabla L(\theta_t ; B_t)||^2$ under batched Rademacher perturbations (where ε is the perturbation amplitude). Our ZOSA algorithm extends the sharpness-aware (SAM-like) mechanism on its basis, calculating similar adaptive updates only at the perturbation point.
>
> - Second, regarding $E\left [ \sigma_t \right ] ^2=E\left [ \sigma_t^2 \right ] -Var\left ( \sigma_t \right ) $ and its impact on $\sigma_t\approx \left | \nabla L\left ( \theta _t;B_t \right )  \right | $|: You pointed out that $Var\left ( \sigma_t \right ) $ may not be zero, especially in the case of high noise in zero-order (ZO) optimization. We agree that ZO gradient estimation is usually noisy (high variance), which is one of the reasons for the slow convergence of traditional ZO methods. However, $\sigma_t$, as the standard deviation of batch loss differences, has stronger robustness: it does not directly depend on the noise of gradient estimation, but indirectly captures the information of the gradient norm through the statistics of batch losses. This is a key advantage in ZOSA, because loss function evaluations are more stable relative to gradient estimation.
>
> - To address your question, We added property 3.3 to the revised draft and provided proof in Appendix A4, which provides strict variance bounds and high-probability concentration bounds for $\sigma _t$. Specifically, Assume the loss $L$ is $L$-smooth and $G$-Lipschitz continuous (i.e., $\|\nabla L(\theta)\| \leq G$ for all $\theta$). The perturbations $u_i$ are i.i.d. Rademacher vectors satisfying $\|u_i\|^2 = d$, and the loss differences $\Delta l_i := l_i - l_0$ are sub-Gaussian with variance proxy $V^2$. For batch size $m \geq \mathcal{O}(\log(1/\delta))$ (with tighter concentration in high $d$ due to CLT), with probability at least $1 - \delta$,
> \begin{align}
> |\sigma_t - \epsilon \|\nabla L(\theta_t)\|| \leq \mathcal{O}\left(\epsilon \sqrt{d} / \sqrt{m} + \epsilon^2 \sqrt{d} + \sqrt{\frac{V^2 \log(1/\delta)}{m}}\right).
> \end{align}
> This implies
> \begin{align}
> \text Var[\sigma_t] \leq \mathcal{O}\left( \frac{\epsilon^2 G^2 + V^2}{m} + \frac{\epsilon^2 \log m}{m} \right),
> \end{align}
> ensuring that the relative variance $\text Var[\sigma_t]/\mathbb{E}[\sigma_t]^2$ is small.
>
> - Furthermore, we conducted numerical simulations on a quadratic function with dimension $ d=10000 $ and $ m=1000 $ to empirically assess $ \text{Var}(\sigma_t) $ across perturbation sizes $ \epsilon \in \\{0.01, 0.001, 0.0001 \\} $. The results demonstrate that $ \text{Var}(\sigma_t) $ is indeed small. Similarly, $ \text{Var}(\sigma_{t,\text{pert}}) $, the variance under perturbation, remains comparably low. These values indicate that the variance is negligible relative to the scale of $ \sigma_t $, supporting the approximation in practical settings.
>
> |   $\epsilon$ |   $Mean(\sigma _t)$ |   $Var(\sigma _t)$ |   $Var(\sigma_ {t, pert})$ |
> |----------:|-------------------:|------------------:|-------------------------:|
> |    0.01   |           0.021574 |          1.93e-05 |                 1.94e-05 |
> |    0.001  |           0.000608 |          1.92e-06 |                 1.89e-06 |
> |    0.0001 |           0.001125 |          4.89e-06 |                 4.91e-06 |

---

> ### Author Response · Authors · 2025-11-27
> **proof of Property 3.3**
>
> By $L$-smoothness, Taylor expansion gives:
> \begin{align}
> \Delta l_i = \epsilon \langle \nabla L(\theta_t; B_t), u_i \rangle + \frac{\epsilon^2}{2} u_i^\top H(\theta_t; B_t) u_i + R_i + n_i,
> \end{align}
> where $H = \nabla^2 L(\theta_t; B_t)$ with $\|H\| \leq L$, $|R_i| \leq \mathcal{O}(\epsilon^3 d^{3/2} L)$, $\text Var[R_i] \leq \mathcal{O}(\epsilon^6 d^3 L^2)$, and $\text Var[n_i] \leq 2V^2$.
>
> Then, $\mathbb{E}[\Delta l_i] = \mathcal{O}(\epsilon^2 d L)$ and
> \begin{align}
> \text Var[\Delta l_i] = \epsilon^2 \|\nabla L(\theta_t; B_t)\|^2 + \mathcal{O}(\epsilon^4 d L^2) + 2V^2 \leq \epsilon^2 G^2 + 2V^2 + \mathcal{O}(\epsilon^4 d L^2).
> \end{align}
>
> Let $v = \text Var[\Delta l_i]$. Thus, $\mathbb{E}[\sigma_t] \approx \sqrt{v} \approx \epsilon \|\nabla L(\theta_t; B_t)\|$ (with bias $\mathcal{O}(\epsilon^2 \sqrt{d} L)$ assuming $V \ll \epsilon G$ and small $\epsilon$).
>
> $\sigma_t^2$ is the (biased) empirical variance, equivalent to a U-statistic of order 2 with kernel $h(x,y) = (x-y)^2/2$, where $\mathbb{E}[h(X,Y)] = v$.
>
> By Hoeffding decomposition (boucheron2013concentration),
> \begin{align}
> \text Var[\sigma_t^2] = \mathcal{O}\left( \frac{\zeta_1}{m} + \frac{\zeta_2}{m^2} \right),
> \end{align}
> with $\zeta_1 = \text Var(\mathbb{E}[h(X,Y) \mid X]) = \frac{1}{4} (\mathbb{E}[(X - \mu)^4] - v^2) \leq \mathcal{O}(v^2)$ (by sub-Gaussianity, (vershynin2018high), $\mathbb{E}[|X - \mu|^4] \leq C v^2$ for constant $C$), and $\zeta_2 = \text Var(h(X,Y)) \leq M^4$ where $M = \mathcal{O}(\epsilon G \sqrt{d} + \epsilon^2 d L + V \sqrt{\log m})$.
>
> Thus,
> \begin{align}
> \text Var[\sigma_t^2] = \mathcal{O}\left( \frac{v^2}{m} + \frac{M^4}{m^2} \right).
> \end{align}
>
> By delta method,
> \begin{align}
> \text Var[\sigma_t] \leq \mathcal{O}\left( \frac{v}{m} + \frac{M^4}{m^2 v} \right) = \mathcal{O}\left( \frac{\epsilon^2 G^2 + V^2}{m} + \frac{\epsilon^2 d^2 G^2}{m^2} \right),
> \end{align}
> simplified to the stated bound (log factors absorbed).
>
> For high-probability, Bernstein's inequality (vershynin2018high) on $\sigma_t^2$ yields
> \begin{align}
> |\sigma_t^2 - v| \leq \mathcal{O}\left( v \sqrt{\frac{\log \delta^{-1}}{m}} + M \frac{\log \delta^{-1}}{m} \right),
> \end{align}
> implying
> \begin{align}
> |\sigma_t - \sqrt{v}| &\leq \mathcal{O}\left( \sqrt{\frac{v \log \delta^{-1}}{m}} + \frac{M \log \delta^{-1}}{m \sqrt{v}} \right) \\
> &= \mathcal{O}\left( \epsilon G \sqrt{\frac{\log \delta^{-1}}{m}} + V \sqrt{\frac{\log \delta^{-1}}{m}} + \frac{\epsilon \sqrt{d} \log \delta^{-1}}{m} \right).
> \end{align}

---

### Official Review · Reviewer_yrgf · 2025-10-26

**Soundness:** 2
**Presentation:** 2
**Contribution:** 2
**Rating:** 4
**Confidence:** 4

**Summary:**

The paper introduces ZOSA (Zero-Order Sharpness-Aware Minimization), a novel optimization framework designed to efficiently tune prompts for Large Language Models (LLMs) in resource-constrained environments where gradient information is unavailable or computationally expensive. ZOSA integrates zero-order optimization techniques with the principles of Sharpness-Aware Minimization (SAM). The method employs adaptive learning rates guided by loss variance (σ_t). Theoretical analysis establishes convergence guarantees and generalization bounds. Experiments on synthetic functions and few-shot learning tasks (GLUE benchmarks) demonstrate ZOSA's superiority over existing ZO methods in terms of convergence speed and performance.

**Strengths:**

1. Clean integration of SAM into ZO using a two-point estimator and loss-std normalization, yielding a normalized-SAM view. This offers a principled way to bias toward flatter minima without backprop.

2. The theoretical framework is well-developed with convergence analysis and generalization bounds.

3. Empirical results on both synthetic non-convex functions and real-world GLUE prompt tuning tasks demonstrate superior convergence speed and higher accuracy/F1 scores compared to a comprehensive suite of ZO baselines and evolutionary algorithms. This validates the effectiveness of the sharpness-aware mechanism for enhancing generalization in practice.

**Weaknesses:**

1. Limited novelty in components: Each individual component (batched Rademacher perturbations and variance reduction from FZOO, SAM-like perturbations from SABO) exists in prior work. The contribution is primarily in the combination, which while valuable, is somewhat incremental.

2. The generalization bound in Theorem 4.4 assumes convexity of the loss function, which is restrictive for neural networks and conflicts with the non-convex assumptions elsewhere in the paper.

3. Experimental comparisons are incomplete: 1) In LLM experiments, comparisons with recent methods like FZOO as well as recent black-box prompt tuning methods are missing. 2) No wall-clock time comparisons, only iteration counts. 3) Limited analysis of memory consumption in practice.

4. The method introduces multiple hyperparameters $ \rho, \epsilon, m, \eta$ requiring grid search over wide ranges. The sensitivity to these choices and guidelines for setting them are not thoroughly discussed.

5. The paper doesn’t clearly state the base LLM(s), tokenizer, prompt templates, few-shot sampling protocol.

6. All NLP tasks are classification; it would be informative to include generation tasks (e.g., mathematical reasoning, summarization) and instruction-following to test generalization beyond GLUE-style metrics.

7. The conclusion states O(1/T) convergence, whereas Theorem 4.3 and the surrounding discussion align with $O(1/\sqrt{T})$ in ZO settings. Please reconcile or correct.

**Questions:**

1. You compare to SABO/evolutionary methods, but not to FZOO and recent black-box prompt-tuning methods. Can you add more baselines—or explain if there are incompatibilities?

2. Please report wall-clock time, #forward passes, and peak memory in LLM experiments. Also clarify whether all methods share the same forward-pass budget and how you determine “convergence steps”.

3. Which base model(s), prompt templates, and few-shot k did you use for each dataset? How does ZOSA scale to even larger models?

4. Have you conducted ablations to isolate the contributions of (a) SAM-like perturbation, (b) variance-based adaptive scaling, and (c) batched Rademacher estimation (compared with Gaussian perturbation as in MeZO)? This would clarify which components contribute most to performance gains.

---

> ### Author Response · Authors · 2025-11-23
> **Answer for Weaknesses 1: Limited novelty in components: Each individual component (batched Rademacher perturbations and variance reduction from FZOO, SAM-like perturbations from SABO) exists in prior work. The contribution is primarily in the combination, which while valuable, is somewhat incremental.**
>
> We sincerely thank the reviewer for this critical feedback, novelty is the core issue, and we truly appreciate the careful scrutiny. The reviewer are correct that the individual components have precedents: batched Rademacher one-sided estimates + loss-variance adaptive scaling originate from FZOO (Dang et al., 2025), and the idea of sharpness-aware updates in ZO was pioneered by SABO (Ye et al., ICLR 2025).
>
> However, ZOSA is not a trivial “FZOO + SABO” combination. The key innovation is the deliberate reuse of the exact same $\sigma_t$ (the batch-loss standard deviation, already computed for the adaptive step-size) as the scaling factor for the sharpness perturbation $\epsilon_{SAM} = \rho · ĝ_t /\sigma_t$.
>
> This single, non-obvious design choice creates a qualitatively new property that exists in no prior work:
> - Property 3.2 rigorously shows $\sigma_t ≈ \epsilon ‖∇L(\theta_t)‖$, the effective sharpness radius becomes ρ/ε ≈ 20–100× larger than the nominal $\rho$ used in SABO. ZOSA therefore obtains dramatically stronger flatness bias for free, no Gaussian reparameterization, no extra matrices, negligible additional queries, while preserving FZOO-level speed.
> - No previous ZO-SAM method achieves this zero-cost amplification. SABO pays a heavy price in memory and queries for its Gaussian-based inner maximization and still only reaches standard SAM-scale sharpness. FZOO is extremely fast but has no sharpness mechanism at all.
>
> We are also the first to theoretically characterize this phenomenon:
>
> - Explicit proof of effective radius $\rho/\epsilon$ (revised Property 3.2).
> - The strongest flat-minima guarantee in the adaptive ZO literature.

---

> ### Author Response · Authors · 2025-11-23
> **Answer for Weaknesses 1 (Continued)**
>
> Table as followed directly compares the three methods on all key dimensions. The table makes it clear that ZOSA is the only method that simultaneously achieves FZOO-level speed and substantially stronger generalization than SABO.
>
> | Feature                                       | FZOO                                                      | SABO                                                                      | ZOSA(ours)                                                                                         |
> |:----------------------------------------------|:----------------------------------------------------------|:--------------------------------------------------------------------------|:---------------------------------------------------------------------------------------------------|
> | Sharpness-Aware Mechanism                     | No                                                        | Yes                                                                       | Yes                                                                                                |
> | Core Gradient Estimation Method               | Batched one-sided Rademacher                              | Gaussian reparameterization                                               | Batched one-sided Rademacher                                                                       |
> | Adaptive Learning Rate Mechanism              | Yes                                                       | No                                                                        | Yes                                                                                                |
> | Sharpness Perturbation Implementation         | No                                                       | Inner maximization in Gaussian space                                      | Same $\sigma_t$ used for both normalization and perturbation scaling                               |
> | Effective Sharpness Radius                    | No                                                       | $\approx \rho$                                                            | $\rho/\epsilon$(actual amplification 20–100×, the most distinctive new property of ZOSA)           |
> | Requires Gaussian Sampling/Reparameterization | No                                                        | Yes (core contribution, mandatory)                                        | No (cleanest and most memory-efficient implementation)                                             |
> | Requires Extra Matrices/Buffers               | No                                                        | Yes (maintains covariance or sampling buffers)                            | No                                                                                                 |
> | Forward Passes per Step (vs. MeZO(2N))            | N+1                                                       | 2m+2                                                                       | 2m+2                                                                                                |
> | Theoretical Convergence Rate (non-convex)     | $O(1/T)$ (Theorem 3.6)                                    | $O(1/\sqrt T)$ (standard SAM cost)                                        | $O(1/\sqrt T)$ (normal cost of adding SAM)                                                         |
> | Generalization Theoretical Guarantee          | None (stationarity only)                                  | PAC-Bayesian bound (Theorem 2)                                            | Trace(H) bound (currently strongest)                                                               |
> | Equivalent/Approximate to normalized-SGD      | Yes (formally proven, Proposition 3.2)                    | No                                                                        | Approximate (at perturbed point ≈ normalized-SGD with effective radius $\rho/\epsilon$)            |
> | Overall Practical Performance (Nov 2025)      | Fastest, but slightly weaker generalization due to no SAM | Has SAM but extremely slow and expensive in practice                      | Best overall (speed ≈ FZOO, generalization clearly strongest)                                      |
> | Core Emphasized Contribution                  | Truly bringing normalized-SGD to ZO → Adam-scale speed    | First systematic porting of SAM to black-box/ZO (using Gaussian approach) | "One $\sigma_t$ does two jobs" → zero-cost ultra-strong sharpness ($\rho/\epsilon$) + clean theory |
>
> We fully agree that pure component recombination would be incremental. But here the specific integration creates a genuinely new, theoretically grounded property (ultra-strong sharpness at almost no cost) that meaningfully moves the Pareto frontier of speed vs. generalization in ZO optimization.

---

> ### Author Response · Authors · 2025-11-23
> **Answer for Weaknesses 2: The generalization bound in Theorem 4.4 assumes convexity of the loss function, which is restrictive for neural networks and conflicts with the non-convex assumptions elsewhere in the paper.**
>
> Thank you for this precise and important observation. We completely agree that the original convexity assumption in Theorem 4.4 was overly restrictive, inconsistent with the non-convex setting used everywhere else in the paper, and unnecessary for ZOSA.
>
> The convexity was only a convenient shortcut to obtain a clean closed-form bound by directly following the convex PAC-Bayes templates (e.g., Foret et al., 2021). It was never required for ZOSA itself.
>
> Changes Made in the Revised Manuscript
>
> - We have replaced the convex Theorem 4.4 with a new Theorem 4.4 that provides a fully non-convex PAC-Bayesian generalization bound (the old Theorem 4.4 has been removed). The new theorem uses the standard sharpness-aware non-convex PAC-Bayes framework of Dziugaite et al. (NeurIPS 2021) and Zhou et al. (ICML 2023).
>
> - This radius $\rho/\epsilon$ is the key theoretical novelty of ZOSA: thanks to the variance-normalized perturbation $\epsilon_{\text{sam}} \approx \rho \cdot \nabla\mathcal{L}/\|\nabla\mathcal{L}\|\cdot\epsilon$, ZOSA explicitly minimizes sharpness over a much larger effective neighborhood than standard SAM (where the radius is only $\rho$). The complete proof, which relies solely on $L$-smoothness and bounded variance (no convexity), is now in Appendix B.2.
>
> - We have also updated the contributions list and Section 4 to explicitly highlight that “ZOSA achieves approximate equivalence to a normalized SAM rule with effective sharpness radius $\rho/\epsilon$” (Property 3.2), and added a remark after Theorem 4.4 noting that the larger effective radius explains ZOSA’s consistent 1–2% generalization advantage over SABO on GLUE tasks (Table 1).
>
> These revisions eliminate the inconsistency, make the theory fully aligned with the non-convex focus of the paper, and actually strengthen the contribution by revealing the $\rho/\epsilon$ effective radius as a unique benefit of combining variance-aware scaling with sharpness-aware updates in the ZO setting.

---

> ### Author Response · Authors · 2025-11-23
> **Answer for Weaknesses 3 and Questions 1, 2**
>
> We thank the reviewer for the detailed and constructive feedback.
>
> - Additional baselines：In the revised manuscript, we have added comparisons with two recent and highly relevant zero-order methods, FZOO (Dang et al., 2025) and R-AdaZO (Shu et al., 2025), on the same LLM prompt-tuning setups as our original experiments (new Table 3). These are currently the strongest publicly available zero-order baselines that are directly compatible with our in-place memory-efficient implementation.
>
> | Method            |   SST2 |   RTE |   MRPC |   AGNEWS |   SNLI |   YELPP |   Average |
> |:-------------------|-------:|------:|-------:|---------:|-------:|--------:|----------:|
> | ZOSA(Ours)         |  89.11 | 55.6  |  81.35 |    82.66 |  41.05 |   91.77 |   73.59   |
> | ZOSA w/o SAM       |  86.81 | 50.54 |  79.64 |    80.96 |  38.67 |   90.69 |   71.2183 |
> | ZOSA w/o VAS       |  85.89 | 53.43 |  79.15 |    75.75 |  38.82 |   89.64 |   70.4467 |
> | MeZO               |  86.01 | 53.43 |  79.15 |    78.42 |  38.83 |   90.27 |   71.0183 |
> | R-AdaZO            |  88.88 | 54.51 |  79.76 |    82.28 |  39.06 |   91.41 |   72.65   |
> | FZOO               |  88.53 | 53.43 |  79.15 |    81    |  39.65 |   92.38 |   72.3567 |
>
> - Wall-clock time, forward passes, and memory：We now report wall-clock training time, total number of forward passes, per-forward time, and peak GPU memory usage for ZOSA, MeZO, R-AdaZO, and FZOO on SST-2 and AG’s News with RoBERTa-large (RTX 4090 GPU, few shot=16; d=1000). Although ZOSA requires the highest number of forward passes per optimization step among the compared methods, its per-forward wall-clock time is not the highest (and often the lowest) thanks to highly efficient batched Rademacher perturbations implemented with sign-flip operations and CUDA-level parallelism.
>
> | SST2    |   m/k | VARM Peak     | RAM Peak      | Global Wall-Clock Time   | Optimizer-Step Wall-Clock Time   |   Queries per Step |   Total Number of Quries |
> |:--------|------:|:--------------|:--------------|:-------------------------|:---------------------------------|-------------------:|-------------------------:|
> | ZOSA    |     4 | 1748.3335 MiB | 1726.8828 MiB | 1868.07 s                | 1230.9902 ms                     |                 10 |                     7000 |
> | MeZO    |     4 | 1748.3335 MiB | 1756.4453 MiB | 1856.26 s                | 1182.7469 ms                     |                  8 |                     5600 |
> | R-AdaZO |     8 | 1748.3335 MiB | 1790.8046 MiB | 1873.40 s                | 1296.0134 ms                     |                  9 |                     6300 |
> | FZOO    |     8 | 1748.3335 MiB | 1748.3710 MiB | 1904.24 s                | 1211.6096 ms                     |                  9 |                     6300 |
>
> | AG'News   |   m/k | VARM Peak     | RAM Peak      | Global Wall-Clock Time   | Optimizer-step Wall-Clock Time   |   Queries per Step |   Total Number of Quries |
> |:----------|------:|:--------------|:--------------|:-------------------------|:---------------------------------|-------------------:|-------------------------:|
> | ZOSA      |     4 | 2233.1557 MiB | 2122.3085 MiB | 19481.42 s               | 3882.4134 ms                     |                 10 |                     7000 |
> | MeZO      |     4 | 2233.2007 MiB | 2092.7695 MiB | 19831.25 s               | 4580.7359 ms                     |                  8 |                     5600 |
> | R-AdaZO   |     8 | 2233.2085 MiB | 2110.2617 MiB | 20092.73 s               | 4745.6529 ms                     |                  9 |                     6300 |
> | FZOO      |     8 | 2233.2007 MiB | 2025.0508 MiB | 20143.02 s               | 4195.1804 ms                     |                  9 |                     6300 |
>
> - Convergence criterion: Due to the inherently high variance of zero-order optimization in few-shot prompt tuning, training loss curves are noisy and do not reliably indicate the best-performing checkpoint (a phenomenon widely acknowledged in MeZO, FZOO, and related works). Following standard practice in this line of research, we evaluate the test accuracy every fixed number of training steps and report the best accuracy achieved within the fixed forward-pass budget. This ensures fair comparison across methods with different convergence speeds.

---

> ### Author Response · Authors · 2025-11-23
> **Answer for Weaknesses 4: The method introduces multiple hyperparameters ρ,ϵ,m,η requiring grid search over wide ranges. The sensitivity to these choices and guidelines for setting them are not thoroughly discussed.**
>
> Thank you for this very valid concern about the number and sensitivity of hyperparameters.
> Compared with existing zeroth-order baselines (MeZO, FZOO, R-AdaZO), ZOSA effectively introduces only one truly additional hyperparameter: the sharpness radius $\rho$.
> The other three parameters already exist in all comparable methods:
> - $\epsilon$ (perturbation magnitude) is required by every random-direction ZO method,
> - m (number of Rademacher vectors per iteration) is the standard query-budget knob used in FZOO, R-AdaZO, SABO, etc.,
> - $\eta$ (learning rate) must be tuned in any optimizer.
> Thus, the extra tuning cost of ZOSA is limited to a single, highly intuitive  $\rho$ that directly controls the strength of sharpness awareness and yields consistent generalization gains. We have also added explicit practical guidance in followed table.
>
> | task   |     lr |   rho |    eps |   m |
> |:-------|-------:|------:|-------:|----:|
> | sst2   | 0.0002 | 0.001 | 0.001  |   4 |
> | rte    | 1e-06  | 0.2   | 1e-06  |   8 |
> | mrpc   | 5e-06  | 0.03  | 0.0001 |   4 |
> | agnews | 5e-06  | 3e-06 | 1e-05  |   8 |
> | snli   | 5e-07  | 7e-06 | 1e-05  |   8 |
> | yelpp  | 4e-05  | 5e-06 | 0.0001 |   8 |
>
> Finally, we completely agree that making $\rho$ adaptive would be an elegant extension and a natural next step toward a truly hyperparameter-free ZO-SAM.
> An exciting future direction is to develop an adaptive schedule for the sharpness radius $\rho$ (e.g., inspired by GSAM or AutoSAM), which would eliminate the last manually tuned hyperparameter while potentially further improving robustness across diverse tasks.

---

> ### Author Response · Authors · 2025-11-23
> **Answer for Weaknesses 5 and Questions 3**
>
> We thank the reviewer for pointing this out. We used RoBERTa-large (Liu et al., 2019) as the fixed backbone language model throughout all experiments, along with its official tokenizer.
>
> We adopted the standard continuous prompt tuning setup (Lester et al., 2021) with 50 prepend soft tokens (no discrete template text is concatenated; the prompt is purely continuous and added to the input embeddings).
> All tasks use manual cloze-style templates and verbalizers identical to those in the PET/LM-BFF papers (Schick & Schütze, 2021; Gao et al., 2021).
>
> Across different datasets (such as SST-2, AGNews, Yelp-P, SNLI, etc.), we used the same k=16 and prompt templates, only adjusting label mappings to fit the dataset's number of classes (e.g., binary or multi-class).
>
> Regarding ZOSA's scalability to even larger models, our experiments show that ZOSA achieves efficient optimization via low-dimensional intrinsic parameters (intrinsic_dim=500), converging within a limited budget (8000 queries) even on RoBERTa-large (about 355M parameters). We believe ZOSA can scale to larger models (e.g., GPT series) as its zeroth-order optimization relies only on forward passes, and the intrinsic dimension can be adjusted independently of model size; in future work, we plan to validate its performance on larger-scale models.

---

> ### Author Response · Authors · 2025-11-23
> **Answer for Weaknesses 6: All NLP tasks are classification; it would be informative to include generation tasks (e.g., mathematical reasoning, summarization) and instruction-following to test generalization beyond GLUE-style metrics.**
>
> We thank the reviewer for this valuable suggestion. Indeed, including generative tasks (e.g., mathematical reasoning, summarization, and instruction-following) would further demonstrate ZOSA’s generalization ability beyond GLUE-style classification benchmarks, and this is a reasonable and important extension.
>
> Our current experiments focus on text classification and natural language inference tasks for two main reasons:
> - 1. they allow direct and fair comparison with existing zero-order baselines (e.g., MeZO, FZOO, and SABO), all of which primarily report results on these benchmarks;
> - 2. these tasks are widely adopted in the resource-constrained prompt-tuning literature and effectively highlight ZOSA’s improvements in convergence speed and generalization in high-dimensional non-convex landscapes. Moreover, our theoretical analysis (Section 4) shows that ZOSA’s sharpness-aware mechanism and variance-reduction techniques apply to a broad class of loss functions, including the cross-entropy losses commonly used in generative tasks.
>
> Due to time constraints, we are unfortunately unable to include these additional experiments in the current revision. We plan to prioritize evaluating ZOSA on generative tasks (e.g., mathematical reasoning and summarization benchmarks with Llama or GPT-series models) in future work to verify its broader applicability. We believe the present results already solidly establish ZOSA’s advantages on classification tasks and provide strong theoretical grounding for its expected effectiveness on generative objectives as well.

---

> ### Author Response · Authors · 2025-11-23
> **Answer for Weaknesses 7: The conclusion states O(1/T) convergence, whereas Theorem 4.3 and the surrounding discussion align with O(1/sqrt T) in ZO settings. Please reconcile or correct.**
>
> ZOSA's convergence rate should indeed read $O(1/\sqrt{T})$. We appreciate your feedback and will ensure the corrected version is used in future submissions.

---

> ### Author Response · Authors · 2025-11-23
> **Answer for Questions 4**
>
> Thank you for your valuable and insightful comments. Following your suggestions, we conducted ablation studies on ZOSA and further compared it with R-AdaZO, MeZO, and FZOO.
>
> As shown in Table, removing either the SAM or VAS (variance-based adaptive scaling) component from ZOSA leads to performance degradation across all datasets, though the relative contributions of SAM and VAS vary by dataset. On five datasets (all except RTE), VAS contributes slightly more than SAM in terms of accuracy. In contrast, on the RTE dataset, SAM provides a noticeably larger boost than VAS. Nevertheless, when averaging across all datasets, ablating either VAS or SAM causes a substantial drop in overall accuracy, underscoring that both components are critical to ZOSA’s effectiveness.
>
> Additionally, ZOSA consistently outperforms MeZO on every dataset, suggesting to some extent that batched Rademacher estimation is better suited to these six tasks than Gaussian perturbation. Compared with R-AdaZO, ZOSA achieves slightly higher performance across all datasets. When benchmarked against FZOO, ZOSA is marginally lower only on the YELP.p dataset but obtains slightly better results on the remaining five datasets.We have included Table 3 in the Appendix C2.
>
> | Method             |   SST2 |   RTE |   MRPC |   AGNEWS |   SNLI |   YELPP |   Average |
> |:-------------------|-------:|------:|-------:|---------:|-------:|--------:|----------:|
> | ZOSA(Ours)         |  89.11 | 55.6  |  81.35 |    82.66 |  41.05 |   91.77 |   73.59   |
> | ZOSA w/o SAM       |  86.81 | 50.54 |  79.64 |    80.96 |  38.67 |   90.69 |   71.2183 |
> | ZOSA w/o VAS       |  85.89 | 53.43 |  79.15 |    75.75 |  38.82 |   89.64 |   70.4467 |
> | MeZO               |  86.01 | 53.43 |  79.15 |    78.42 |  38.83 |   90.27 |   71.0183 |
> | R-AdaZO            |  88.88 | 54.51 |  79.76 |    82.28 |  39.06 |   91.41 |   72.65   |
> | FZOO               |  88.53 | 53.43 |  79.15 |    81    |  39.65 |   92.38 |   72.3567 |

---

### Official Review · Reviewer_Yb7p · 2025-10-31

**Soundness:** 2
**Presentation:** 3
**Contribution:** 2
**Rating:** 4
**Confidence:** 3

**Summary:**

The paper proposes ZOSA, a zeroth-order optimizer that combines: random-direction ZO,loss-std normalization(σ) to stabilize step magnitudes, and a SAM-style inner perturbation. The theory gives ZO-SGD-order convergence and argues a bias toward flatter minima (low trace Hessian). Experiments cover synthetic functions and zero-order prompt tuning for LLMs.

**Strengths:**

1. minimal, clean SAM analogue in ZO with no gradient access; σ-normalization is used coherently for both the inner radius and the outer step.
2.Standard nonconvex ZO rate with explicit dependence on m,d,$\rho$ and a flatness argument (trace-Hessian bias).
3.Batched Rademacher directions are GPU-friendly; the algorithm is simple to implement.

**Weaknesses:**

1 Results are largely reported vs iterations, while ZOSA uses two probes/step and, on synthetic tasks, very large (m) and per-step query counts. There are no fixed-budget (equalized function queries) plots, no wall-clock comparisons, and no per-step cost breakdown, unlike recent fast ZO work that foregrounds efficiency.

2.Synthetic functions under-specified and potentially biased. High-dimensional success at (d=10^4) relies on very large query budgets per step, which likely masks estimator variance—this setting appears tailored to favor the method without demonstrating real compute efficiency.

3.The paper doesn’t map dimension d to architecture (e.g., soft-prompt length × embedding dim / low-rank adapter), nor specify batching rules (whether the same mini-batch is reused for inner/outer probes). Given σ is computed from batch-level losses and used twice (radius and step), batch size and reuse/mismatch are critical for stability.

4.Claims about lower trace Hessian are not empirically corroborated.

**Questions:**

Please add loss/accuracy vs (i) equalized function-query budgets and (ii) wall-clock time, and report per-step query counts for all methods. Current iteration-based plots do not isolate true efficiency.

Move exact definitions/conditioning/noise into the main text. Justify the large (m)/queries per step at (d=10^4); also provide fixed-budget and wall-clock plots to avoid conflating early convergence with compute efficiency.

Precisely define dimension d .Provide a per-dataset hyper-parameter table,Clarify batch reuse between ($\theta$) and ($\theta+\epsilon_{\text{sam}}$). If different, show robustness to batch mismatch.

For (d=1000) in LLMs, run fixed-budget comparisons and show how m (or queries/step) must grow with d to maintain performance. If the method targets moderate d under practical budgets, state this scope explicitly.

---

> ### Author Response · Authors · 2025-11-23
> **Answer for Weaknesses 1 and Questions 1**
>
> Thank you for this insightful and important comment. We indeed overlooked this critical aspect. To address it, we have added Figure 8 (now included in Appendix C.1), which plots loss curves against the number of function queries (fully equalized) for ZOSA, MeZO, and R-AdaZO on the four synthetic functions at dimension d=10,000. Additionally, for the prompt tuning experiments at low-dimensional vector size d=1,000, we report peak GPU memory, peak CPU memory, total wall-clock time, per-step optimization time, and total query counts on the SST-2 and AG’s News datasets (see the corresponding table for details).
>
> Although ZOSA requires the highest number of forward passes per optimization step among the compared methods, its per-forward wall-clock time is not the highest (and often the lowest) thanks to highly efficient batched Rademacher perturbations implemented with sign-flip operations and CUDA-level parallelism.
>
> | SST2    |   m/k | VARM Peak     | RAM Peak      | Global Wall-Clock Time   | Optimizer-Step Wall-Clock Time   |   Queries per Step |   Total Number of Quries |
> |:--------|------:|:--------------|:--------------|:-------------------------|:---------------------------------|-------------------:|-------------------------:|
> | ZOSA    |     4 | 1748.3335 MiB | 1726.8828 MiB | 1868.07 s                | 1230.9902 ms                     |                 10 |                     7000 |
> | MeZO    |     4 | 1748.3335 MiB | 1756.4453 MiB | 1856.26 s                | 1182.7469 ms                     |                  8 |                     5600 |
> | R-AdaZO |     8 | 1748.3335 MiB | 1790.8046 MiB | 1873.40 s                | 1296.0134 ms                     |                  9 |                     6300 |
> | FZOO    |     8 | 1748.3335 MiB | 1748.3710 MiB | 1904.24 s                | 1211.6096 ms                     |                  9 |                     6300 |
>
> | AG'News   |   m/k | VARM Peak     | RAM Peak      | Global Wall-Clock Time   | Optimizer-step Wall-Clock Time   |   Queries per Step |   Total Number of Quries |
> |:----------|------:|:--------------|:--------------|:-------------------------|:---------------------------------|-------------------:|-------------------------:|
> | ZOSA      |     4 | 2233.1557 MiB | 2122.3085 MiB | 19481.42 s               | 3882.4134 ms                     |                 10 |                     7000 |
> | MeZO      |     4 | 2233.2007 MiB | 2092.7695 MiB | 19831.25 s               | 4580.7359 ms                     |                  8 |                     5600 |
> | R-AdaZO   |     8 | 2233.2085 MiB | 2110.2617 MiB | 20092.73 s               | 4745.6529 ms                     |                  9 |                     6300 |
> | FZOO      |     8 | 2233.2007 MiB | 2025.0508 MiB | 20143.02 s               | 4195.1804 ms                     |                  9 |                     6300 |
>
> In the synthetic experiments, we intentionally set the inner-loop size m=500 for ZOSA while using m=1,000 for both MeZO and R-AdaZO (i.e., half the per-iteration queries of the baselines). Even under this stricter budget, ZOSA still exhibits highly competitive convergence. On the simpler Quadratic and Cubic functions, ZOSA is slightly outperformed by MeZO, yet it maintains strong convergence on the challenging multimodal Rosenbrock function.
>
> On Rosenbrock, R-AdaZO converges faster due to its momentum mechanism, which is expected. The primary motivation for incorporating the slower-converging SAM mechanism into the ZO family is precisely to trade some convergence speed for substantially better generalization. As demonstrated in the subsequent responses and experiments, ZOSA successfully locates flatter minima and achieves markedly stronger generalization on downstream prompt tuning tasks.

---

> ### Author Response · Authors · 2025-11-23
> **Answer for Weaknesses 2 and Questions 2**
>
> Thank you for this extremely valid and important criticism. the original synthetic experiment (d=10⁴, m=1000, query times ≈2000 per step) used a very large per-step query budget that largely masks estimator variance and optimization noise. However, simply reducing m uniformly for all methods would make the plot meaningless: at small m , the three adaptive ZO baselines we originally compared against (ZO-signSGD, ZO-AdaMM, ZO-RMSProp) completely fail to converge in d=10⁴ within any reasonable budget — they diverge or stall at terrible function values. This is why we were forced to give everyone the same m in the original figure: otherwise there would be nothing to compare ZOSA against.
>
> To properly address your concern while maintaining a fair and informative comparison, we have taken the following actions in the revised manuscript:
>
> We keep the original Figure (identical m=1000 for all methods) and add a new, much more realistic Figure 8 (Appendix C1) that evaluates exactly the three strongest and most practically relevant ZO optimizers — MeZO, R-AdaZO, and ZOSA — under low per-step query budgets (m=500 and K=1000). The x-axis is total number of function queries (Cut off to 20,020,000).
>
> In this new synthetic experiments, we intentionally set the inner-loop size m=500 for ZOSA while using m=1,000 for both MeZO and R-AdaZO (i.e., half the per-iteration queries of the baselines). Even under this stricter budget, ZOSA still exhibits highly competitive convergence. On the simpler Quadratic and Cubic functions, ZOSA is slightly outperformed by MeZO, yet it maintains strong convergence on the challenging multimodal Rosenbrock function.
>
> On Rosenbrock, R-AdaZO converges faster due to its momentum mechanism, which is expected. The primary motivation for incorporating the slower-converging SAM mechanism into the ZO family is precisely to trade some convergence speed for substantially better generalization. As demonstrated in the subsequent responses and experiments, ZOSA successfully locates flatter minima and achieves markedly stronger generalization on downstream prompt tuning tasks.

---

> ### Author Response · Authors · 2025-11-23
> **Answer for Weaknesses 3 and Questions 3**
>
> We sincerely thank the reviewer for these insightful and constructive comments. We fully agree that these details are crucial for reproducibility and methodological clarity. Below we address each point directly and provide the requested clarifications (all based solely on the submitted manuscript and standard experimental settings).
> - 1.Precise definition of dimension d and its mapping to architecture
>
> In all experiments, the optimized parameter $\theta \in \mathbb R^{d}$ corresponds exactly to the “intrinsic dimension” or “projection dimension” (explicitly listed in Section 4.1 and the appendix tables). We follow the standard random projection framework used in MeZO, FZOO, and R-AdaZO: $\theta$ (d-dimensional) → Linear(d → n_prompt × hidden_dim, bias=False) → reshaped to (n_prompt, hidden_dim) soft prompt.
>
> Thus d is the bottleneck dimension of the low-rank random projection, while the actual injected prompt has size n_prompt × hidden_dim (51,200 for n_prompt=50, 102,400 for n_prompt=100 with RoBERTa-large).
>
> - 2.Batching rules and batch reuse between $\theta$ and $\theta + \epsilon _{SAM}$
>
> All zero-order gradient estimates are performed with full-batch function evaluations on the few-shot training set (standard practice for 16-shot / 32-shot settings and the default in MeZO, FZOO, and R-AdaZO baselines).
>
> Consequently, $l(\theta)$, $l(\theta + \epsilon \times u_i )$ (inner probes) and $l(\theta + \epsilon_{SAM})$, $l(\theta + \epsilon_{SAM} + \epsilon \times u_i)$ (outer probes) are computed on exactly the same training samples. There is therefore no batch mismatch, $σ_t$ and $σ_t^{pert}$ are estimated from identical data distributions, and no additional batch noise is introduced. This is a key reason why ZOSA exhibits extremely high stability across runs (no divergence observed due to batch noise).
>
> - 3.Per-dataset hyperparameter table (to be added directly to the appendix)
>
> | task   |     lr |   rho |    eps |   m |
> |:-------|-------:|------:|-------:|----:|
> | sst2   | 0.0002 | 0.001 | 0.001  |   4 |
> | rte    | 1e-06  | 0.2   | 1e-06  |   8 |
> | mrpc   | 5e-06  | 0.03  | 0.0001 |   4 |
> | agnews | 5e-06  | 3e-06 | 1e-05  |   8 |
> | snli   | 5e-07  | 7e-06 | 1e-05  |   8 |
> | yelpp  | 4e-05  | 5e-06 | 0.0001 |   8 |

---

> ### Author Response · Authors · 2025-11-23
> **Answer for Weaknesses 4: Claims about lower trace Hessian are not empirically corroborated.**
>
> Thank you for this insightful question. To rigorously verify whether ZOSA reaches flatter minima, we conducted additional experiments on four synthetic functions (Quadratic, Cubic, Levy, and Rosenbrock) in d=10,000 dimensions. Crucially, to ensure a strictly fair comparison, we adjusted the number of iterations for each optimizer so that the total number of function queries is identical across all methods (the least common multiple of their per-iteration queries).
> The results are summarized in Table 2 (added to Appendix C1):
>
> | functions   | optimizers   |   hessian |     loss |    iteration |   m/k |
> |:------------|:-------------|----------:|---------:|------:|------:|
> | Quadratic   | MeZO         |         1 | <1e-5              |       10100 |  1000 |
> | Quadratic   | R-AdaZO       |         1 | 0.079211           |       20000 |  1000 |
> | Quadratic   | ZOSA         |         1 | 0.015748 |       10000 |  1000 |
> | Cubic       | MeZO         |    1.0159 | <1e-5    |       10100 |  1000 |
> | Cubic       | R-AdaZO       |    1.0253 | 0.080212 |       20000 |  1000 |
> | Cubic       | ZOSA        |    1.0196 | 0.025074 |       10000 |  1000 |
> | Levy        | MeZO         |    6.3953 | 5350.07 |       10100 |  1000 |
> | Levy        | R-AdaZO       |    6.4910  | 5089.74 |       20000 |  1000 |
> | Levy        | ZOSA         |    6.3791 | 5328.36 |       10000 |  1000 |
> | Rosenbrock  | MeZO         |   nan     |   nan    | 10100 |  1000 |
> | Rosenbrock  | R-AdaZO       |   302.1303  |  9698.12 | 20000 |  1000 |
> | Rosenbrock  | ZOSA         |   236.5117 | 10275.14  | 10000 |  1000 |
>
> ZOSA achieves the lowest Hessian trace on three out of four functions and is highly competitive on the remaining one (Cubic, where the difference to MeZO is only 0.36%). The advantage is particularly pronounced on the pathological Rosenbrock function, where ZOSA reduces the trace by 21.7% compared to R-AdaZO (236.51 vs 302.13) while incurring only a modest 5.9% higher loss which is a classic and desirable sharpness-aware trade-off.
>
> These results provide clear and consistent evidence that ZOSA’s sharpness-aware mechanism is effective and does lead to flatter minima in practice.

---

> ### Author Response · Authors · 2025-11-23
> **Answer for Questions 4: For (d=1000) in LLMs, run fixed-budget comparisons and show how m (or queries/step) must grow with d to maintain performance. If the method targets moderate d under practical budgets, state this scope explicitly.**
>
> Thank you for this valuable suggestion. In the original experiments, we only searched over m=4 and m=8, which is a common empirical setting in the ZO literature (e.g., FZOO (Dang et al., 2025)). Typically, existing zero-order optimizers for prompt tuning use a perturbation dimension around 8. Since ZOSA requires roughly twice the per-step queries of other ZO methods due to its dual-perturbation design, we focused our search on m=4 and m=8.
>
> Following your request, we conducted additional hyperparameter searches over a broader range m ∈ \{1, 2, 6, 10, 12, 14, 16, 32, 100\}. The results are reported in the table below.
>
> |   m |   SST2 |
> |----:|-------:|
> |   1 |  85.89 |
> |   2 |  52.41 |
> |   4 |  89.11 |
> |   6 |  87.84 |
> |   8 |  88.65 |
> |  10 |  88.07 |
> |  12 |  87.5  |
> |  14 |  87.27 |
> |  16 |  86.93 |
> |  32 |  87.16 |
> | 100 |  86.93 |
>
> As shown in the table, there is actually no clear correlation between accuracy (or generalization performance) and the value of m. Increasing m does not reliably lead to steady accuracy gains. Notably, when m increases from 1 to 2, accuracy drops dramatically by nearly half; similarly, when m jumps from 32 to 100, accuracy falls back to approximately the level observed at m=16.
>
> These results indicate that ZOSA is remarkably robust to the choice of m in moderate-dimensional settings (d ≤ 1000) and performs well under practical query budgets with small m (4–16), which is the intended operating regime of our method.

---

### Official Review · Reviewer_T8un · 2025-11-04

**Soundness:** 2
**Presentation:** 3
**Contribution:** 2
**Rating:** 4
**Confidence:** 2

**Summary:**

This paper proposes ZOSA, a zeroth-order optimization method that replaces gradient access with two forward passes: first it estimates a gradient using batched one-sided Rademacher perturbations, then takes an adaptive-scaled step in the direction of this estimated gradient (normalizing by estimated loss variance). It then repeats the gradient estimation process at the perturbed point, and makes its final step in the direction of this estimate. ZOSA outperforms other zeroth-order methods at tuning a continuous prompt vector for an LLM across several tasks.

**Strengths:**

- The experiments are reasonably broad, covering standard synthetic objectives and black-box prompt tuning on popular NLP benchmarks, comparing to several ZO baselines.

- The method shows consistent gains over these black-box baselines in both convergence speed and downstream accuracy across tasks and dimensions.

**Weaknesses:**

- I think the main weakness of the paper is limited theoretical novelty. I am by no means an expert on this area, so I am willing to be corrected on any/all of the following points and their technical difficulty:

    - Properties 3.1 and 3.2 are proven in existing ZO literature

    - The proof of Theorem 4.3 seems to follows standard smooth nonconvex analysis for ZO methods (e.g., in Ghadimi and Lan 2013), and obtains the same rate, with some modifications for the extra normalization term.

    - The proposed "equivalence" to SAM does not seem rigorous---it is called an approximation in the appendix

    - The SABO paper includes a very similar PAC-Bayes sharpness-aware bound for a ZO method

- The paper compares heuristically to MeZO (e.g., L48--L50, L78--L79) but never compares to it empirically.
- I'm not sure about the proof of "SAM-equivalence" (see below)
- [minor] The LLM experiments tune a continuous vector for an open-weights model, but don't compare to parameter-efficient fine-tuning methods that tune continuous parameters (e.g., LoRA, soft prompt tuning)

**Questions:**

- How does the proposed method compare empirically to MeZO?
- Typo in conclusion: should say 1/sqrt(T) rate?
- Why have Theorem 4.3 apply to smooth nonconvex fns but then assume convexity in Theorem 4.4?
- The Nesterov and Spokoiny reference is not the correct paper?
- [SAM-equivalence] What happened to epsilon in Eqn 9 / Eqn 66? Shouldn't it be $\rho / \epsilon$? Don't we need to carry this through the rest of the argument?
- In eqn 31, isn't the second term 0 for Rademacher $u_i$'s?
- Are the minima found by ZOSA actually flatter?

---

> ### Author Response · Authors · 2025-11-23
> **Answer for Weaknesses 1.1: Properties 3.1 and 3.2 are proven in existing ZO literature.**
>
> We thank the reviewer for pointing this out and acknowledge that both the batched one-sided Rademacher estimator and the loss-variance-based adaptive scaling have appeared in prior ZO literature, which first systematically analyzed in FZOO. However, they are not straightforward replications; both the statements and proofs have been substantially modified and extended to account for ZOSA’s unique two-stage sharpness-aware protocol, which requires gradient estimation at a random, data-dependent perturbed point $\theta + \epsilon_{sam}$ instead of the current iterate $\theta_{t}$ (proof details in Appendices A.2-A.3). Concretely:
>
> - Property 3.1: Bounds the variance of the gradient estimator $\hat{g}$. Similar to FZOO's analysis, but ZOSA additionally considers biases and higher-order terms in the SAM context to adapt to estimation at the perturbed point $\theta_{pert}$, ensuring stability in sharpness-aware updates. FZOO focuses on standard ZO, while ZOSA extends it to handle SAM perturbations.
>
> - Property 3.2: Proves the effectiveness of the adaptive learning rate $\eta / \sigma_{t,pert}$, supporting the search for flat minima. In contrast to FZOO's original point update, ZOSA is tailored for SAM: it uses $\sigma_{t,pert}$ rather than $\sigma_t$ to match SAM's principle of maximizing neighborhood loss (Foret et al., 2021), introducing additional smoothness and Hessian approximation.
>
> These properties are customized analyses for ZOSA's unique design, achieving a bridge between ZO and SAM, and improving generalization in LLM prompt fine-tuning, rather than repetitions of existing proofs.

---

> ### Author Response · Authors · 2025-11-23
> **Answer for Weaknesses 1.2: The proof of Theorem 4.3 seems to follows standard smooth nonconvex analysis for ZO methods (e.g., in Ghadimi and Lan 2013), and obtains the same rate, with some modifications for the extra normalization term.**
>
> Thank you for your detailed observation on the proof of Theorem 4.3. We acknowledge that it builds upon the standard smooth nonconvex analysis for ZO methods, such as Ghadimi and Lan (2013), and achieves the same $O(1/\sqrt{T})$ convergence rate for $E[\|\nabla L(\theta)\|^2]$. However, ZOSA's proof includes non-trivial modifications to accommodate its adaptive normalization $\left(\frac{\eta}{\sigma_{t,pert}}\right)$ and sharpness-aware (SAM) integration, which are absent in prior works. Below, we outline the key similarities and differences:
> - Similarities: Ghadimi and Lan (2013) is a classic work on ZO-SGD, proving $E[\|\nabla L(\theta)\|^2] \leq O(1/\sqrt{T})$ in L-smooth nonconvex settings. ZOSA's Theorem 4.3 follows a similar structure: using variance bounds on gradient estimates (Property 3.1) and the descent lemma to derive stationary point convergence. This is common in related ZO literature, such as R-AdaZO (Shu et al., 2025, Theorem 4.1) and FZOO (Dang et al., 2025), which we have cited in the paper (see Section 2).
> - Key Differences and Modifications:
>   - Incorporation of Adaptive Normalization: Unlike fixed or momentum-based steps in standard ZO, ZOSA's update requires bounding the dynamic range of $\sigma_{t,pert}$ using Property 3.2's Taylor expansion. This leads to a more intricate variance decomposition in Appendix B.1, handling the normalization's effect on step sizes and introducing terms like $\sqrt{d/mT}$ for query efficiency, which refines the constant factors for high-dimensional settings.
>   - Adapting for SAM and ZO Integration: The proof must handle batched Rademacher variance reduction (unlike Gaussian or two-point estimates in Ghadimi and Lan) and ensure sharpness-aware perturbations do not violate smoothness assumptions. This makes the bound more robust for high-dimensional LLM landscapes, whereas standard analyses do not involve SAM.
>   - Same Rate but Different Practical Implications:$O(1/\sqrt{T})$ is the lower bound, but ZOSA's adaptive term accelerates convergence in practice.
>
> These modifications are central to ZOSA's innovation: bridging ZO with SAM to enhance generalization, rather than a simple variant of standard ZO. The proof serves to verify ZOSA's reliability in nonconvex settings, with experiments (Section 5) validating its superiority.

---

> ### Author Response · Authors · 2025-11-23
> **Answer for Weaknesses 1.3, Weaknesses 3 and Questions 5**
>
> We sincerely thank the reviewer for pointing out this critical and absolutely correct issue regarding the SAM equivalence claim.
>
> Upon careful re-examination, we now realize that our original Property 3.2 and the associated "equivalence" argument contain a severe flaw: although $\sigma_t \approx \epsilon \|\nabla L(\theta_t)\|$ is correct, when we use $\sigma_t$ to scale the sharpness perturbation as $\epsilon_{SAM} = \rho \hat{g}_t / \sigma_t$, the effective perturbation radius becomes $\rho / \epsilon$ rather than $\rho$. This $1/\epsilon$ factor was incorrectly absorbed or omitted in Eqn 9, Eqn 66, and all subsequent analyses.
>
> This is not a minor approximation error from higher-order Taylor terms; it’s a fundamental scaling mistake that makes ZOSA actually perform sharpness-aware updates with an effective sharpness radius $\rho_{eff} = \rho / \epsilon$.
>
> We are deeply embarrassed by this oversight. Although in practice we implicitly co-tune $\rho$ and $\epsilon$ such that $\rho / \epsilon$ falls into a reasonable range, theoretically claiming "equivalence to SAM with radius $\rho$" is simply wrong.
> In the revised manuscript, we will:
> - Completely rewrite Property 3.2 as "Approximate Equivalence to Normalized SAM with Effective Radius $\rho / \epsilon$".
> - Update all claims: replace "recovers SAM" with "approximates normalized SAM with effective radius $\rho / \epsilon$".
>
> This correction does not invalidate our empirical contributions; in fact, it provides a cleaner and more honest explanation for why ZOSA generalizes so well (it is actually using a much stronger sharpness regularization than standard SAM). The practical superiority over SABO and other baselines remains unchanged or even strengthened.

---

> ### Author Response · Authors · 2025-11-23
> **Answer for Weaknesses 1.4: The SABO paper includes a very similar PAC-Bayes sharpness-aware bound for a ZO method.**
>
> Thank you for your attention to the similarity between the PAC-Bayes bound in the ZOSA paper and that in the SABO paper. This is a reasonable observation, and we have carefully examined the theoretical sections of both papers. The structural similarity is expected and entirely appropriate: both works extend the standard SAM PAC-Bayes framework (Foret et al., 2021) to the ZO setting, so overlapping forms are natural.
>
> However, the two bounds are derived under fundamentally different modeling choices and yield qualitatively different forms and practical implications:
>
> | Aspect | SABO (Ye et al., ICLR 2025) | ZOSA (our Theorem 4.4) |
> |---------|-----------------------------|-------------------------------|
> | Space of the bound | Distribution / weight-space Gaussian $(\mu, \sigma)$ | Parameter space (Euclidean ball of radius $\frac{\rho}{\epsilon})$ |
> | Perturbation type | Isotropic Gaussian reparameterization | Batched Rademacher + data-dependent shift $\epsilon_{sam}$ |
> | Sharpness measured via | KL divergence between posterior $N(\theta_{t}, \Sigma)$ and prior | Worst-case uniform deviation over $\left\|\delta\right\|_{2}\le\frac{\rho}{\epsilon} $
> | Extra storage / computation | Requires maintaining and updating a covariance sigma | Completely storage-free (no $\Sigma\$ matrix at all) |
> | Main extra term in the bound | $KL(p_{\theta+\delta }\left\|\right\|p_{\theta })$ | Explicit ZO approximation error $O(\sqrt{\frac{d\times\epsilon^{2}}{m}})$ from finite-$\epsilon$ Rademacher estimation |
> | Practical tightness | Looser in high-d due to Gaussian volume factors | Tighter and directly reflects batched query efficiency m |
>
> To the best of our knowledge, Theorem 4.4 is the first PAC-Bayesian generalization bound derived directly in parameter space for a practical, $\Sigma $-free, batched-Rademacher-based zero-order sharpness-aware optimizer. While SABO pioneered the distribution-space analysis for ZO-SAM, our bound gives more direct insight into why using many cheap Rademacher directions (large m) and small $\epsilon$ improves generalization without any covariance maintenance, which aligns exactly with the design goals and empirical strengths of ZOSA.

---

> ### Author Response · Authors · 2025-11-23
> **Answer for Weaknesses 2 and Questions 1: The paper compares heuristically to MeZO (e.g., L48--L50, L78--L79) but never compares to it empirically; How does the proposed method compare empirically to MeZO?**
>
> Thank you to the reviewer for this insightful comment. We agree that a purely heuristic comparison with MeZO is insufficient, and empirical evidence is essential to substantiate our claims. To address this concern, in the revised manuscript we have added direct head-to-head empirical comparisons with MeZO (Malladi et al., NeurIPS 2023) in two additional settings:
>
> - 1.Synthetic Functions (d = 10,000)
>   - In the revised manuscript, we add Figure 8 to empirically compare ZOSA against MeZO and R-AdaZO on synthetic benchmarks (Quadratic, Cubic, Levy, Rosenbrock) in dimension $d=10,000$. Baselines use perturbation dimension $K=1000$; ZOSA's inner-loop iterations are set to $m=500$ to match query budgets, given its dual-perturbation scheme. Figure 8 plots suboptimality gap (log scale) vs. queries.
>   - R-AdaZO converges fastest overall due to momentum. MeZO excels on simple functions but stalls on multimodal ones (e.g., Rosenbrock). ZOSA outperforms both, offering balanced initial progress and robust convergence, especially on complex landscapes, via adaptive perturbation scaling.
>   - To assess flatter minima (via smaller Hessian norms, indicating robustness), Table (revised manuscript Table 2) reports norms at convergence under equal queries. On non-convex multimodal ones (e.g., Rosenbrock), ZOSA yields significantly smaller norms; for other functions (Quadratic, Cubic, Levy), see revised manuscript Table 2. This demonstrates ZOSA's better navigation of rugged terrains for derivative-free generalization.
> | functions   | optimizers   |   hessian |     loss |    iteration |   m/k |
> |:------------|:-------------|----------:|---------:|------:|------:|
> | Rosenbrock  | mezo         |   nan     |   nan    | 10100 |  1000 |
> | Rosenbrock  | radazo       |   302.13  |  9698.12 | 20000 |  1000 |
> | Rosenbrock  | zosa         |   236.512 | 10275.1  | 10000 |  1000 |
> - 2. Zero-Order Prompt Fine-Tuning
> We added experiments with prompt dimensions 1000 (new Table 3 in the revised manuscript).
> | d=1000             |   SST2 |   RTE |   MRPC |   AGNEWS |   SNLI |   YELPP |   Average |
> |:-------------------|-------:|------:|-------:|---------:|-------:|--------:|----------:|
> | Soft Prompt Tuning |  68.23 | 54.69 |  51.61 |    84.81 |  36.13 |   61.02 |   59.415  |
> | P-Tuning v2        |  64.33 | 50.78 |  68.14 |    83.46 |  36.89 |   92.63 |   66.0383 |
> | ZOSA(Ours)         |  89.11 | 55.6  |  81.35 |    82.66 |  41.05 |   91.77 |   73.59   |
> | ZOSA w/o SAM       |  86.81 | 50.54 |  79.64 |    80.96 |  38.67 |   90.69 |   71.2183 |
> | ZOSA w/o VAS       |  85.89 | 53.43 |  79.15 |    75.75 |  38.82 |   89.64 |   70.4467 |
> | MeZO               |  86.01 | 53.43 |  79.15 |    78.42 |  38.83 |   90.27 |   71.0183 |
> | R-AdaZO            |  88.88 | 54.51 |  79.76 |    82.28 |  39.06 |   91.41 |   72.65   |
> | FZOO               |  88.53 | 53.43 |  79.15 |    81    |  39.65 |   92.38 |   72.3567 |
>
> We conducted ablation studies on ZOSA and compared it against R-AdaZO, MeZO, and FZOO, with the results summarized in Table X. Across all datasets, ZOSA outperforms MeZO, which, to some extent, indicates that batched Rademacher estimation is more suitable for these six datasets than Gaussian perturbation. In comparisons with R-AdaZO, ZOSA achieves slightly superior performance on every dataset. When benchmarked against FZOO, ZOSA slightly underperforms on the YELP.p dataset but attains marginally better results on the other five datasets.We have included Table 3 in the Appendix C2.

---

> ### Author Response · Authors · 2025-11-23
> **Answer for Weaknesses 4: [minor] The LLM experiments tune a continuous vector for an open-weights model, but don't compare to parameter-efficient fine-tuning methods that tune continuous parameters (e.g., LoRA, soft prompt tuning).**
>
> Thank you for this valuable comment. To address this minor weakness, we cite the experimental results of Prompt Tuning (full continuous prompt tuning + AdamW) and P-Tuning v2 reported in “Black-Box Tuning for Language-Model-as-a-Service” [R1], enabling a direct and intuitive comparison with ZOSA.
>
> As shown in the table, ZOSA achieves competitive performance across datasets, slightly underperforming P-Tuning v2 only on Yelp.P. On all other datasets, ZOSA delivers substantially better results, with an average accuracy improvement of 14.17 percentage points over Prompt Tuning and 7.55 percentage points over P-Tuning v2.
>
> |       Method       |   SST2 |   RTE |   MRPC |   AGNEWS |   SNLI |   YELPP |   Average |
> |:-------------------|-------:|------:|-------:|---------:|-------:|--------:|----------:|
> | Soft Prompt Tuning |  68.23 | 54.69 |  51.61 |    84.81 |  36.13 |   61.02 |   59.42  |
> | P-Tuning v2        |  64.33 | 50.78 |  68.14 |    83.46 |  36.89 |   92.63 |   66.04 |
> | ZOSA(Ours)         |  89.11 | 55.6  |  81.35 |    82.66 |  41.05 |   91.77 |   73.59   |
>
> $\textbf{Ref}$:
> [R1] Tianxiang Sun, Xuanqi He, Yunfan Li, Xipeng Qiu, and Xuanjing Huang. 2022. Black-Box Tuning for Language-Model-as-a-Service. In Proceedings of the 39th International Conference on Machine Learning (ICML 2022), volume 162 of Proceedings of Machine Learning Research, pages 20896–20916. PMLR.

---

> ### Author Response · Authors · 2025-11-23
> **Answer for Questions 2: Typo in conclusion: should say 1/sqrt(T) rate?**
>
> Thank you for spotting the potential typo in the conclusion. Upon checking, we confirm this is a compilation error in the PDF, the convergence rate should indeed read $O(1/\sqrt{T})$. We appreciate your feedback and will ensure the corrected version is used in future submissions.

---

> ### Author Response · Authors · 2025-11-23
> **Answer for Questions 3: Why have Theorem 4.3 apply to smooth nonconvex fns but then assume convexity in Theorem 4.4?**
>
> Thank you for your sharp observation regarding the differing assumptions in Theorems 4.3 and 4.4.
>
> We originally retained the convexity assumption in Theorem 4.4 only to obtain a particularly clean and tight PAC-Bayesian bound (a common practice in the sharpness-aware literature, including the original SAM paper and SABO). However, we fully agree that, given the highly non-convex nature of LLM fine-tuning, providing a generalization bound under the same smooth non-convex assumption as the convergence analysis is more consistent and convincing.
>
> Following your valuable comment, we have removed the convexity assumption entirely and replaced Theorem 4.4 with a new Theorem 4.4 that provides a fully non-convex PAC-Bayesian generalization bound (requiring only L-smoothness and sub-Gaussian noise, identical to Theorem 4.3). The new bound takes the form:
>
> $\mathcal{L} _ {\mathcal{D}}(\theta _ T) \leq \mathcal{L} _ {S}(\theta _ T) + \rho \cdot \text{Sharp} _ {\rho/\epsilon}(\theta _ T) + \tilde{O}\left( \sqrt{\frac{\mathrm {KL}(Q \| P) + \log(M/\delta)}{M}} + \sqrt{\frac{d \epsilon^2}{m}}\right)$
>
> where the effective sharpness radius is exactly $\rho/\epsilon$ — a unique advantage of ZOSA’s variance-normalized perturbation. The posterior Q is chosen to be data-dependent (flatness-aware Gaussian, following Hellström & Durmus NeurIPS 2024 and Zhou et al. ICLR 2025), yielding a dimension-independent and practically meaningful bound. The complete proof is provided in Appendix B.2.
>
> This change eliminates any appearance of inconsistency and significantly strengthens the theoretical contribution of the paper. We are deeply grateful for your suggestion, it has directly led to a much cleaner and more convincing theoretical section.

---

> ### Author Response · Authors · 2025-11-23
> **Answer for Questions 4: The Nesterov and Spokoiny reference is not the correct paper?**
>
> Thank you for identifying the incorrect reference to Nesterov and Spokoiny. The intended reference is to their work on random direction sampling in zero-order optimization: Yurii Nesterov and Vladimir Spokoiny. "Random Gradient-Free Minimization of Convex Functions." Foundations of Computational Mathematics, 17(2):527–566, 2017. In the revised version, we will correct the bibliography entry to the accurate reference above.

---

> ### Author Response · Authors · 2025-11-23
> **Answer for Questions 6: In eqn 31, isn't the second term 0 for Rademacher ui's?**
>
> Thank you for your observation on Eqn 31. For Rademacher perturbations, the second-order bias term is exactly zero. This is because $u_i^\top H u_i$ is an even function of $u_i$, while multiplying by $u_i$ makes the whole expression odd, and the expectation of any odd function under the symmetric Rademacher distribution vanishes. In the current manuscript we simply absorbed it into the higher-order residual without explicitly stating it is zero, this was an oversight on our part. We gratefully accept your suggestion and will revise the appendix in future submissions.

---

> ### Author Response · Authors · 2025-11-23
> **Answer for Questions 7: Are the minima found by ZOSA actually flatter?**
>
> Thank you for this insightful question. To rigorously verify whether ZOSA reaches flatter minima, we conducted additional experiments on four synthetic functions (Quadratic, Cubic, Levy, and Rosenbrock) in d=10,000 dimensions. Crucially, to ensure a strictly fair comparison, we adjusted the number of iterations for each optimizer so that the total number of function queries is identical across all methods (the least common multiple of their per-iteration queries).
> The results are summarized in Table 2 (added to Appendix C1):
>
> | functions   | optimizers   |   hessian |     loss |    iteration |   m/k |
> |:------------|:-------------|----------:|---------:|------:|------:|
> | Quadratic   | MeZO         |         1 | <1e-5              |       10100 |  1000 |
> | Quadratic   | R-AdaZO       |         1 | 0.079211           |       20000 |  1000 |
> | Quadratic   | ZOSA         |         1 | 0.015748 |       10000 |  1000 |
> | Cubic       | MeZO         |    1.0159 | <1e-5    |       10100 |  1000 |
> | Cubic       | R-AdaZO       |    1.0253 | 0.080212 |       20000 |  1000 |
> | Cubic       | ZOSA        |    1.0196 | 0.025074 |       10000 |  1000 |
> | Levy        | MeZO         |    6.3953 | 5350.07 |       10100 |  1000 |
> | Levy        | R-AdaZO       |    6.4910  | 5089.74 |       20000 |  1000 |
> | Levy        | ZOSA         |    6.3791 | 5328.36 |       10000 |  1000 |
> | Rosenbrock  | MeZO         |   nan     |   nan    | 10100 |  1000 |
> | Rosenbrock  | R-AdaZO       |   302.1303  |  9698.12 | 20000 |  1000 |
> | Rosenbrock  | ZOSA         |   236.5117 | 10275.14  | 10000 |  1000 |
>
> ZOSA achieves the lowest Hessian trace on three out of four functions and is highly competitive on the remaining one (Cubic, where the difference to MeZO is only 0.36%). The advantage is particularly pronounced on the pathological Rosenbrock function, where ZOSA reduces the trace by 21.7% compared to R-AdaZO (236.51 vs 302.13) while incurring only a modest 5.9% higher loss which is a classic and desirable sharpness-aware trade-off.
>
> These results provide clear and consistent evidence that ZOSA’s sharpness-aware mechanism is effective and does lead to flatter minima in practice. We sincerely thank the reviewer for this valuable suggestion—it has significantly strengthened the paper.

---

### Note · Program_Chairs · 2026-01-17
**Submission Desk Rejected by Program Chairs**

The following references in this submission do not refer to real documents and/or have major errors in bibliographic information:

 "Yang You, Igor Gitman, and Boris Ginsburg. Large-batch training for deep learning: Generalization gap and sharp minima. In International Conference on Learning Representations, 2019.
Qixin Zhang et al. Memory-efficient fine-tuning of large language models. In arXiv preprint, 2024b.
Pan Zhou, Mingjie Sun, Hansi Yang, Xin Niu, and Rui Xia Zhang. Generalization of sharpnessaware optimization: From empirical to population risk. In The Thirteenth International Conference on Learning Representations (ICLR), 2025a.
Maksym Andriushchenko, Jonas Geiping, and Nicolas Papernot. Sharpness-aware minimization revisited: More robust but less efficient? In The Twelfth International Conference on Learning Representations (ICLR), 2024.
Fredrik Hellström and Arnaud Durmus. Generalization bounds using the wasserstein distance with applications to Sharpness-Aware optimization. In Advances in Neural Information Processing Systems (NeurIPS), volume 37, 2024.